

# The importance of vertical resolution in the free troposphere for modeling intercontinental plumes

Jiawei Zhuang[1], Daniel J. Jacob[1], Sebastian D. Eastham[2]

[1]School of Engineering and Applied Sciences, Harvard University, Cambridge, MA 02138, USA
[2]Laboratory for Aviation and the Environment Department of Aeronautics and Astronautics, Massachusetts Institute of Technology, Cambridge, MA 02139, USA

*Correspondence to*: Jiawei Zhuang (jiaweizhuang@g.harvard.edu)

**Abstract.** Chemical plumes in the free troposphere can preserve their identity for more than a week as they are transported on intercontinental scales. Current global models cannot reproduce this transport. The plumes dissipate too rapidly due to numerical diffusion in sheared flow. We demonstrate how model accuracy can be limited by either horizontal resolution ($\Delta x$) or vertical resolution ($\Delta z$). Balancing horizontal and vertical numerical diffusion, and weighing by computational cost, implies

an optimal grid resolution ratio $(\Delta x/\Delta z)_{opt} \sim 1000$ for plume transport. This is considerably higher than current global models ($\Delta x/\Delta z \sim 20$) and explains the rapid plume dissipation in these models as limited by vertical resolution. Plume simulations with the GFDL-FV3 global dynamical core over a range of horizontal and vertical grid resolutions confirm this limiting behavior. Our highest-resolution simulation ($\Delta x \approx 25$ km, $\Delta z \approx 80$ m) preserves the maximum volume mixing ratio in the plume to within 35% after 8 days in strongly sheared flow, a drastic improvement over current models. The local surface pollution influence

from the subsiding plume on intercontinental scales is also considerably increased. Adding free tropospheric vertical levels in global models is computationally inexpensive and would also improve the simulation of water vapor.

## 1 Introduction

Global transport of pollution mainly takes place in the free troposphere where winds are strong and pollutant lifetimes are long. The free troposphere extends from the top of the planetary boundary layer (PBL, typically 2 km altitude) up to the

tropopause. It is a moderately stable environment with strong wind shear. Much of pollution transport in the free troposphere takes place as plumes, typically ~1 km thick in the vertical, that fan out horizontally over a ~1000 km scale and may preserve their coherent structure for up to 1-2 weeks (Crawford et al., 2004; Heald et al., 2003; Liang et al., 2007; Newell et al., 1999; Stoller et al., 1999; Thouret et al., 2000). Global Eulerian models dissipate these plumes too rapidly, due to numerical diffusion introduced by the advection schemes. Although the high-order advection schemes used in these models are highly accurate

under uniform flows, the accuracy breaks down in realistic sheared/stretched flows where plumes filament and the ability to resolve cross-plume gradients is rapidly compromised (Rastigejev et al., 2010). Eastham and Jacob (2017) found that increasing the horizontal resolution of models to address this problem is only of marginal benefit and suggested that the main limitation is vertical resolution. Here we use GFDL-FV3, a global 3-D dynamical core that explicitly solves atmospheric





dynamic equations, to understand the horizontal and vertical resolution requirements for models to simulate global-scale plume transport.

Preserving chemical plumes during global-scale transport is important for representing non-linear chemical and aerosol
processes (Wild and Prather, 2006) and for quantifying intercontinental influences on surface air (Lin et al., 2012; Zhang et al., 2014). For example, models are unable to capture the thick plumes of Asian ozone pollution frequently observed at 2-5 km altitude over California (Hudman et al., 2004; Nowak et al., 2004). Air quality agencies in California have claimed that they cannot meet the current surface ozone standard because of this Asian pollution influence (Neuman et al., 2012). Models find Asian pollution influence in surface air over California to be only a few ppb (Goldstein et al., 2004; Zhang et al., 2008), but
since they cannot resolve the structure of Asian pollution plumes crossing the Pacific they have little credibility.

General Circulation Models (GCMs) used for global simulations of atmospheric dynamics including meteorological data assimilation have increased their resolutions 1000-fold over the past 50 years, buoyed by the growth of computing power (Balaji, 2015). Increasing horizontal resolution has been privileged, and attention to vertical resolution has mainly focused on
the PBL. For example, assimilated meteorological data produced operationally by the NASA Goddard Earth Observing System (GEOS) started in the 1990s with 2°×2.5° horizontal resolution and 20 vertical levels (GEOS-1; Schubert et al. 1993). Today the operational GEOS forward processing (GEOS-FP) product uses a cubed-sphere C720 horizontal resolution (≈ 0.125°) and 72 vertical levels (Lucchesi, 2017). This represents a 20-fold increase in horizontal resolution but only a 4-fold increase in vertical resolution. In the free troposphere at 2-10 km altitude the vertical resolution has increased by only a factor of 2 from
GEOS-1 (8 levels) to GEOS-FP (15 levels). In NOAA' s Next Generation Global Prediction System (NGGPS) program, several state-of-science dynamical cores are tested at horizontal resolutions of 12 km and 3 km, but only with 128 vertical layers -- not trying to improve on the current generation of models (Michalakes et al., 2015). ECMWF's Integrated Forecasting System (IFS) increased its horizontal resolution from 16 km to 9 km in 2016 but its vertical resolution remains at 137 levels (Haiden et al., 2016). On the Sunway TaihuLight supercomputer, a dynamical core is tested at an unprecedentedly high global
horizontal resolution of 488 m but with only 128 vertical layers (Yang et al., 2016).

There are important reasons why horizontal resolution is a priority in GCMs, as reviewed by Haarsma et al. (2016). Increasing horizontal resolution improves the simulation of large-scale features such as the El Niño–Southern Oscillation (ENSO), as well as small-scale features such as tropical cyclones. It has been argued that increasing vertical resolution should follow suit,
based on the ability to resolve fronts and gravity waves. Pecnick and Keyser (1989) recommend an optimal relationship between horizontal and vertical grid spacing

$$\left(\frac{\Delta x}{\Delta z}\right)_{opt} = \frac{1}{s} \; , \tag{1}$$





where $s$ is the frontal slope, and $\Delta x$ and $\Delta z$ are the horizontal and vertical grid spacings respectively. $s$ typically ranges from 0.005 to 0.02 for synoptic-scale fronts, so the optimal $\Delta x/\Delta z$ would be in the range 50-200. Lindzen and Fox-Rabinovitz (1989) recommend

$$\left(\frac{\Delta x}{\Delta z}\right)_{opt} = \frac{N}{f} \quad , \tag{2}$$

where $N$ is the Brunt–Väisälä frequency and $f$ is the Coriolis parameter. $N/f \sim 100$ is Prandtl's ratio, measuring the ratio between the horizontal and vertical scales of geostrophic flows (Dritschel and McKiver, 2015). Based on both Eq. (1) and Eq. (2), $\Delta x/\Delta z$ in GCMs is recommended to be of the order of 100 (Chapter 3.2.1 of Warner, 2010). The current generation of GCMs with $\Delta x \sim 10$ km and $\Delta z \sim 0.5$ km (thus $\Delta x/\Delta z \sim 20$) in the free troposphere is beginning to fall outside that range.

Preserving chemical plumes may have its own requirements of $\Delta x/\Delta z$ related to typical plume sizes and numerical diffusion. In idealized tests by Kent et al. (2012), where plumes were advected by a solid-body rotation flow coupled with vertical oscillation, doubling the vertical resolution brought down the numerical diffusion error by more than half. Numerical diffusion is considerably more severe in realistic sheared/stretched atmospheric flows (Rastigejev et al., 2010). Eastham and Jacob (2017) used GEOS-FP meteorological data to drive the off-line GEOS-Chem Chemical Transport Model (CTM) with

horizontal resolutions ranging from 0.25°×0.3125° (native) to 4°×5°, all with the native 72-level vertical resolution of GEOS-FP. They found that increasing the horizontal resolution is effective in preserving 2-D (horizontal) plumes, but fails with 3-D plumes because the coarse vertical resolution of the native GEOS-FP data is the limiting factor. They could not increase the vertical resolution in the GEOS-FP environment and thus could not explore the issue further.

Increasing free tropospheric vertical resolution in GCMs would seem to be an important consideration for resolving the transport of water vapor, similar to chemical plumes (Pope et al., 2001; Tompkins and Emanuel, 2000). Water vapor in the free troposphere is layered in the same way as other chemical species (Newell et al., 1999; Thouret et al., 2000). An early intercomparison of GCMs found that the radiative effect of water vapor is relatively insensitive to model vertical resolution (Ingram, 2002), which might explain the lack of attention to this issue. However, all GCMs in that intercomparison had coarse

resolution that would make them inadequate for addressing the problem properly.

Here we use the GFDL-FV3 dynamical core as a computationally flexible framework to explore the horizontal and vertical resolution requirements for free tropospheric plume transport. The dynamical core solves the atmospheric dynamics equations with no complications from physical parameterizations. In a full GCM, one would need to account for the vertical resolution

dependence of physical parameterization schemes (Kent et al., 2012; Lane et al., 2000). In a dry dynamical core, we are free to choose any horizontal and vertical resolutions to solve the dynamics equations. A realistic sheared/stretched atmospheric flow can be simulated in a dry dynamical core by triggering baroclinic instability (Jablonowski and Williamson 2006).



## 2 Theoretical analysis

### 2.1 Numerical diffusion and its relation to grid resolution

Numerical diffusion in a Eulerian chemical transport model is caused by numerically solving the 3-D advection equation:

$$\frac{\partial C}{\partial t} + u\frac{\partial C}{\partial x} + v\frac{\partial C}{\partial y} + w\frac{\partial C}{\partial z} = 0, \tag{3}$$

where $C$ is the volume mixing ratio (VMR) and $(u, v, w)$ are the horizontal and vertical wind components. Numerical schemes in models typically use high-order approximations to the upstream derivatives, and often use the flux form of the advection equation (based on number density) to facilitate mass conservation. A first-order scheme allows here a simple analysis, and is relevant to our problem because higher-order schemes degrade to first order as a plume gets stretched to be resolved by only a few grid cells (Huynh, 1997; Rastigejev et al., 2010). The VMR form allows us to focus on numerical diffusion, since the

true solution of the advection equation translates VMR downwind while conserving its magnitude even in divergent flow (Chapter 7.2 of Brasseur and Jacob, 2017).

Using a 3-D first-order upwind scheme with no cross terms, applied to a grid cell $(i, j, k)$ with time level $n$ and wind vector components $(u, v, w)$ all positive, Eq. (3) is approximated by:

$$\frac{C(n+1,i,j,k)-C(n,i,j,k)}{\Delta t} + u\frac{C(n,i,j,k)-C(n,i-1,j,k)}{\Delta x} + v\frac{C(n,i,j,k)-C(n,i,j-1,k)}{\Delta y} + w\frac{C(n,i,j,k)-C(n,i,j,k-1)}{\Delta z} = 0. \tag{4}$$

Apply the Taylor expansion to each term in Eq. (4), for example

$$C(n, i-1, j, k) = C(n, i, j, k) - \Delta x\frac{\partial C}{\partial x} + \frac{\Delta x^2}{2}\frac{\partial^2 C}{\partial x^2} + o(\Delta x^2). \tag{5}$$

which yields for that term

$$u\frac{\partial C}{\partial x} - u\frac{C(n,i,j,k)-C(n,i-1,j,k)}{\Delta x} = u\frac{\Delta x}{2}\frac{\partial^2 C}{\partial x^2} + o(\Delta x). \tag{6}$$

The right-hand side of Eq. (6) is the truncation error between the $u(\partial C/\partial x)$ term in the true equation Eq. (3) and its numerical approximation in Eq. (4). Adding up the error for each term in Eq. (3), we obtain the total truncation error ε:

$$\varepsilon = -\frac{\Delta t}{2}\frac{\partial^2 C}{\partial t^2} + \frac{u\Delta x}{2}\frac{\partial^2 C}{\partial x^2} + \frac{v\Delta y}{2}\frac{\partial^2 C}{\partial y^2} + \frac{w\Delta z}{2}\frac{\partial^2 C}{\partial z^2} + o(\Delta t + \Delta x + \Delta y + \Delta z). \tag{7}$$

In typical truncation error analysis, terms like $\Delta x$ are important because their order indicates the order of accuracy of the scheme, while terms like $\partial^2 C/\partial x^2$ are just coefficients. The scheme here is first-order because the error decreases linearly with





$\Delta x$. The Modified Equation Approach (Chapter 3.3.2 of Durran, 2010; Warming and Hyett, 1974) provides a different view. We can modify the advection equation (Eq. 3) to add the error terms from Eq. (7) on the right-hand side:

$$\frac{\partial C}{\partial t} + u\frac{\partial C}{\partial x} + v\frac{\partial C}{\partial y} + w\frac{\partial C}{\partial z} = -\frac{\Delta t}{2}\frac{\partial^2 C}{\partial t^2} + \frac{u\Delta x}{2}\frac{\partial^2 C}{\partial x^2} + \frac{v\Delta y}{2}\frac{\partial^2 C}{\partial y^2} + \frac{w\Delta z}{2}\frac{\partial^2 C}{\partial z^2}. \tag{8}$$

Using the same scheme (Eq. 4) to solve this modified equation, the error becomes second-order, i.e. decreases quadratically ($\Delta x^2$):

$$\varepsilon' = o(\Delta t + \Delta x + \Delta y + \Delta z). \tag{9}$$

Thus, we can say that, instead of representing the original advection equation (Eq. 3), the numerical scheme (Eq. 4) better
represents the advection-diffusion equation (Eq. 8) with the diffusion term

$$D = -\frac{\Delta t}{2}\frac{\partial^2 C}{\partial t^2} + \frac{u\Delta x}{2}\frac{\partial^2 C}{\partial x^2} + \frac{v\Delta y}{2}\frac{\partial^2 C}{\partial y^2} + \frac{w\Delta z}{2}\frac{\partial^2 C}{\partial z^2}. \tag{10}$$

This view is different from Eq. (7) in that terms like $\partial^2 C/\partial x^2$ can now be interpreted as explicit diffusion in the differential equation while terms like $\Delta x$ become just coefficients. The magnitude of the diffusion term decreases as the resolution
increases, i.e., as the grid spacing $\Delta x$ decreases, bringing the numerical scheme closer to the original equation (Eq. 3).

The time derivative $(\Delta t/2)\partial^2 C/\partial t^2$ in Eq. (10) is not a standard diffusion term and does not have a clear physical meaning, but we can show following Odman (1997) that in the 1-D upwind scheme it is approximated by the spatial derivative (see Appendix A for proof):

$$\frac{\Delta t}{2}\frac{\partial^2 C}{\partial t^2} \approx \alpha\frac{u\Delta x}{2}\frac{\partial^2 C}{\partial x^2}, \tag{11}$$

where $\alpha = u\Delta t/\Delta x$ is the Courant number with value between 0 and 1 (CFL condition). This means, as long as the CFL condition is satisfied, that the time discretization error will not be larger than the spatial discretization error and will not limit the overall accuracy. We omit it in what follows and only consider

$$D \approx \frac{u\Delta x}{2}\frac{\partial^2 C}{\partial x^2} + \frac{v\Delta y}{2}\frac{\partial^2 C}{\partial y^2} + \frac{w\Delta z}{2}\frac{\partial^2 C}{\partial z^2}. \tag{12}$$

If the horizontal grid spacings ($\Delta x$, $\Delta y$) decrease while the vertical grid spacing ($\Delta z$) remains the same, we will eventually reach a point where

$$\left|\frac{u\Delta x}{2}\frac{\partial^2 C}{\partial x^2}\right| + \left|\frac{v\Delta y}{2}\frac{\partial^2 C}{\partial y^2}\right| \ll \left|\frac{w\Delta z}{2}\frac{\partial^2 C}{\partial z^2}\right|, \tag{13}$$



which implies

$$D \approx \frac{w\Delta z}{2} \frac{\partial^2 C}{\partial z^2},$$ (14)

Under this condition, the numerical diffusion is independent of the horizontal resolution and only depends on the vertical resolution. This explains why Eastham and Jacob (2017) found that increasing the horizontal resolution beyond 1°×1° in their model did not lead to further reduction in plume dissipation. Similarly, if the vertical resolution increases, the numerical diffusion will eventually be determined by the horizontal resolution.

**2.2 Balancing horizontal and vertical numerical diffusion**

To avoid being limited by one dimension, the horizontal and vertical diffusion in Eq. (12) should have similar magnitude:

$$\left| \frac{u\Delta x}{2} \frac{\partial^2 C}{\partial x^2} \right| + \left| \frac{v\Delta y}{2} \frac{\partial^2 C}{\partial y^2} \right| \approx \left| \frac{w\Delta z}{2} \frac{\partial^2 C}{\partial z^2} \right|.$$ (15)

The horizontal grid spacings ($\Delta x$, $\Delta y$), wind velocities ($u$, $v$), and plume sharpness ($\partial^2 C/\partial x^2$, $\partial^2 C/\partial y^2$) are all similar in atmospheric applications, thus we have

$$\left| \frac{u\Delta x}{2} \frac{\partial^2 C}{\partial x^2} \right| \approx \left| \frac{v\Delta y}{2} \frac{\partial^2 C}{\partial y^2} \right|.$$ (16)

The optimal grid spacing can then be obtained by equating horizontal and vertical diffusion:

$$2\left| \frac{u\Delta x}{2} \frac{\partial^2 C}{\partial x^2} \right| = \left| \frac{w\Delta z}{2} \frac{\partial^2 C}{\partial z^2} \right|.$$ (17)

Rearranging, we obtain an expression for the optimal ratio between horizontal and vertical grid resolution to balance the effect of numerical diffusion:

$$\left( \frac{\Delta x}{\Delta z} \right)_{opt} = \frac{|w\frac{\partial^2 C}{\partial z^2}|}{2|u\frac{\partial^2 C}{\partial x^2}|}.$$ (18)

Let $L$ and $H$ be the horizontal and vertical extents of the plume and $\Delta C$ be the change in VMR from the center of the plume to the background. We have

$$\left| \frac{\partial^2 C}{\partial x^2} \right| \approx \frac{\Delta C}{L^2}, \quad \left| \frac{\partial^2 C}{\partial z^2} \right| \approx \frac{\Delta C}{H^2}.$$ (19)



The above approximation can be obtained by either scale analysis (Chapter 2.4 of Holton, 2004) or a finite-difference approximation at the center of the plume ($x = x_0$):

$$\frac{\partial^2 C}{\partial x^2}\Big|_{x=x_0} \approx \frac{C|_{x=x_0+L}+C|_{x=x_0-L}-2C|_{x=x_0}}{L^2} = -\frac{2\Delta C}{L^2} \quad . \tag{20}$$

Replacing into Eq. (18), we get

$$\left(\frac{\Delta x}{\Delta z}\right)_{opt} = \frac{w}{2u}\left(\frac{L}{H}\right)^2 . \tag{21}$$

Eq. (21) means that the optimal ratio of horizontal and vertical grid resolutions depends on the wind velocity and the plume

aspect ratio. To get an intuition for this, consider two extreme cases: (1) if $w = 0$, i.e. the 3D advection problem degrades to 2D, there will be no vertical diffusion and thus no requirement on the vertical resolution ($\Delta z_{opt} \to \infty$); (2) if $H \to 0$, i.e. the plume is infinitely thin, we will need infinitely small vertical grids to resolve it ($\Delta z_{opt} \to 0$). In the real atmosphere with typical wind speeds $u = 10$ m s$^{-1}$, $w = 1$ cm s$^{-1}$ and typical plume sizes $L = 1000$ km, $H = 1$ km, we get $(\Delta x/\Delta z)_{opt} = 500$, higher than the dynamical criteria reviewed above. Although this numerical value is little more than an order-of-magnitude estimate,

considering the uncertainty in the individual terms, it suggests that numerical diffusion of chemical plumes may place greater restriction on model vertical resolution than atmospheric dynamics. Later in this paper we will use numerical simulations to derive $(\Delta x/\Delta z)_{opt}$ and compare to the result of this theoretical analysis.

### 2.3 Applying computational cost considerations

In practice, the trade-off between horizontal resolution ($\Delta x$) and vertical resolution ($\Delta z$) must be considered in the context of a given allocation of computational resources. Increasing horizontal resolution by a factor $m$ increases the number of grid cells by $m^2$, since the increase is applied to both the $x$ and $y$ dimensions. In addition, the time step must generally be decreased by a factor $m$ to satisfy the CFL condition, so that the computation cost scales as $m^3$. Increasing vertical resolution does not generally affect the CFL condition because vertical winds are weak relative to $\Delta z$. A fixed amount of computation can thus be expressed

by $\Delta z \Delta x^2 = P$ (where $P$ is a constant), ignoring the CFL condition, or by $\Delta z \Delta x^3 = P$, accounting for the CFL condition.

Here we consider the general problem of minimizing the numerical diffusion for a given allocation of computational resources and with a trade-off parameter $k$ where $k = 1$ represents equal costs for decreasing $\Delta x$ and $\Delta z$, $k = 2$ represents a quadratic cost of decreasing $\Delta x$ (because of corresponding decrease in $\Delta y$), and $k = 3$ represents a cubic cost of decreasing $\Delta x$ (factoring in

the CFL condition):





$$\Delta x^k \Delta z = P. \tag{22}$$

From Eq. (12) and (16), the magnitude of the numerical diffusion term can be written as

$$D = A\Delta x + B\Delta z. \tag{23}$$

where $A$ and $B$ are coefficients:

$$A = \left| u \frac{\partial^2 c}{\partial x^2} \right|, B = \left| \frac{w}{2} \frac{\partial^2 c}{\partial z^2} \right|. \tag{24}$$

In Section 2.2 and following Eq. (21) we estimated $B/A \approx 500$ for typical atmospheric conditions.

For a given amount of computing $P$, the optimal $\Delta x/\Delta z$ ratio is the one that minimizes the numerical diffusion term $D$. This minimum is readily found graphically, as illustrated in Fig. 1. In this Figure, the filled contours are isolines of $D$ as given by Eq. (23) with $B/A = 500$. The solid lines are the computational trade-offs $\Delta z \Delta x^2 = P$ and $\Delta z \Delta x^3 = P$. For a given value of $P$, the numerical diffusion is minimized when the contour lines of $D(\Delta x, \Delta z)$ and $P(\Delta x, \Delta z)$ are parallel, i.e., when their gradients have the same direction:

$$\nabla P(\Delta x, \Delta z) \propto \nabla D(\Delta x, \Delta z). \tag{25}$$

From Eq. (22), $\nabla P(\Delta x, \Delta z) = (k\Delta x^{k-1}\Delta z, \Delta x^k) = \Delta x^{k-1} (k\Delta z, \Delta x)$. From Eq. (23), $\nabla D(\Delta x, \Delta z) = (A, B)$. Thus Eq. (25) becomes

$$(k\Delta z, \ \Delta x) \propto (A, \ B), \tag{26}$$

which yields

$$\left( \frac{\Delta x}{\Delta z} \right)_{opt} = \frac{kB}{A}. \tag{27}$$

In Section 2.2 we implicitly assumed that the computational costs of adjusting $\Delta x$ or $\Delta y$ would be the same, i.e., $k = 1$ in Eq. (22). Eq. (27) is then the same as Eq. (18), and using the same estimate $B/A = 500$ as in Section 2.2 yields $(\Delta x/\Delta z)_{opt} = 500$.

Accounting for higher computational cost when increasing horizontal resolution ($k > 1$) results in a higher optimal ratio. The dashed lines in Fig. 1 show the optimal $\Delta x/\Delta y$ ratios derived for $k = 2$ and $k = 3$. For $k = 2$ we find $(\Delta x/\Delta z)_{opt} = 1000$, and for $k = 3$ we find $(\Delta x/\Delta z)_{opt} = 1500$. It is actually remarkable that the dependence of this optimal ratio on $k$ is linear rather than exponential. The reason is that it is based on the relative contributions of numerical diffusion in the horizontal vs. vertical directions; if numerical diffusion is caused by a coarse horizontal grid, then increasing vertical resolution (even if cheap) will

not provide benefit.



## 3 Atmospheric plume simulation in the GFDL-FV3 dynamical core

We conduct an 8-day simulation of a chemically inert plume in the GFDL-FV3 (https://www.gfdl.noaa.gov/fv3/, "FV3" hereinafter) global 3D dynamical core, with realistic sheared/stretched turbulent flow generated through a baroclinic instability test. FV3 uses the cubed-sphere geometry of Putman and Lin (2007) and the vertically-Lagrangian discretization of Lin (2004).

The horizontal tracer transport algorithm is a high-dimension extension of the third-order Piecewise Parabolic Method (PPM) (Lin and Rood, 1996) but is formally second-order accurate due to operator splitting between the two dimensions (Ullrich et al. 2010). The cubed-sphere grid avoids the polar singularity in the regular latitude-longitude grid and therefore permits efficient global high-resolution simulations on massively parallel machines. An intuitive explanation of the cubed-sphere geometry and resolution notation can be found at http://acmg.seas.harvard.edu/geos/cubed_sphere.html. FV3 has been

implemented as a dynamical core in many global models including the NASA Goddard Earth Observing System (GEOS-5), the NCAR Community Earth System Model (CESM), the NCEP Global Forecast System (GFS) and the High-Performance version of GEOS-Chem (GCHP) (see https://www.gfdl.noaa.gov/fv3/fv3-applications/).

Numerical diffusion takes place in FV3 during Eulerian horizontal advection (due to finite differencing of the spatial

derivatives) and during vertical remapping of the Lagrangian surfaces to the model grid (due to interpolation error). Vertical remapping can use a larger time step than horizontal advection, but the interpolation scheme can be very diffusive if monotonicity is required. Our own comparisons of the vertically Lagrangian scheme to a high-order Eulerian scheme show that they have similar vertical diffusion (Appendix B).

An effective way to emulate realistic turbulent atmospheric flows in a dynamical core is the baroclinic instability test, originally developed by Jablonowski and Williamson (2006) as a dynamical core benchmark and subsequently used in tracer transport simulations (Jablonowski et al., 2008; Ullrich et al., 2016). Baroclinic instability is the main mechanism for cyclogenesis in mid-latitudes. Instability can be triggered by applying a small perturbation to an initial reference state in geostrophic and hydrostatic balance. Starting from the initial perturbation, the baroclinic wave typically becomes observable around model day

4 and generates strong cyclones by day 8 (Jablonowski and Williamson 2006).

Here we initialize the baroclinic instability simulation for 8 days so that cyclones become intense enough for realistic flow shearing/stretching. We then place an inert tracer plume of uniform VMR = 1.0 at the location where flow stretching is the strongest. The initial plume extends horizontally and vertically over a number of grid cells depending on the grid resolution,

as detailed below. We continue the simulation for 8 days and diagnose the transport of the plume. Tracer transport uses the same grid resolution as the dynamics.



We conduct simulations at horizontal cubed-sphere resolutions ranging from C48 (≈200 km) to C384 (≈25 km) and vertical resolutions ranging from L20 (20 vertical layers) to L160. The vertical layers are equally spaced in pressure from the surface (1000 hPa in the reference state) to 1 hPa altitude. Thus L20 has a vertical resolution of 50 hPa, corresponding to 0.6 km in the free troposphere at 600 hPa, which is roughly the vertical resolution of the GEOS-FP product used in the current version

of GEOS-Chem. L160 has a vertical resolution of 6 hPa (roughly 80 m in the free troposphere), well beyond the resolution of any of the current global models.

The time step for the Lagrangian remapping is 30 minutes for the lowest horizontal resolution case (C48) and is reduced proportionally at higher horizontal resolutions. Within this time step are 8 sub-steps for horizontal dynamics calculations. The

frequency of horizontal tracer advection calculations is determined on-the-fly based on the CFL criterion.

The plume is initialized with a uniform mixing ratio VMR = 1.0 over a horizontal area corresponding to 6×6 C48 grid squares (roughly 1000 km × 1000 km), and vertically in a single layer in the L20 case (roughly 0.6 km thick) centered at 625 hPa (4 km) altitude. Thus our coarsest simulation C48L20 resolves the initial plume with 6×6×1 grid cells, while our finest simulation

C384L160 resolves it with 48×48×8 grid cells. The initialization is intended to describe a pollution plume once it has been lifted to the free troposphere and undergone fast initial horizontal fanning (Andreae et al., 1988; Heald et al., 2003). CTMs are generally successful at simulating this initial fanning but then fail to preserve the plume during the subsequent intercontinental transport (Heald et al., 2006; Zhang et al., 2008). The sensitivity of our results to the initial plume size will be discussed.

## 4 Results and discussion

### 4.1 Plume transport and stretching

Fig 2 shows the surface pressures and 700 hPa wind fields on Day 8 of the plume simulation, at C48L20 and C384L20 resolutions. The simulation describes a typical quasi-geostrophic system at mid-latitudes with low and high pressure centers and the associated geostrophic winds. We find that increasing the horizontal resolution intensifies the cyclones, as shown in previous studies (Jablonowski and Williamson, 2006; Lauritzen et al., 2010), while increasing vertical resolution from L20 to

L160 has almost no effect. Hence the GCM emphasis on increasing horizontal resolution.

Also shown in Fig 2 is the local Lyapunov exponent $\lambda = \partial u/\partial x$ of the wind field, which defines the rate constant at which nearby air parcels separate in the direction of the flow, i.e., the intensity of flow stretching. (Rastigejev et al., 2010) showed theoretically that the Lyapunov exponent should be a predictor of numerical diffusion in Eulerian models, and Eastham and

Jacob (2017) confirmed this in GEOS-Chem model simulations. We calculate the Lyapunov exponent locally by (Eastham and Jacob 2017):



$$\lambda \approx \frac{|\Delta u| + |\Delta v|}{\Delta x + \Delta y}, \tag{28}$$

where $\Delta u$ and $\Delta v$ are the changes in wind speeds between the local grid cell and the grid cell downwind, and $\Delta x$ and $\Delta y$ are the corresponding grid spacings. Between 30°N and 60°N where the plume transport takes place, the Lyapunov exponents are

of order $10^{-5}\,s^{-1}$, consistent with values derived from the GEOS-FP wind data (Eastham and Jacob, 2017). Higher horizontal resolution increases stretching because small-scale eddies are better resolved, which offsets some of the gains in reducing numerical diffusion (Rastigejev et al., 2010). We find a mean 700 hPa vertical wind speed $w$ at 30°-60°N of $-0.1 \pm 1.0$ cm s$^{-1}$ ($\pm$ one standard deviation), typical of the range of vertical wind speeds in the real atmosphere. Thus the FV3 simulation provides a realistic environment to investigate how global-scale transport of chemical plumes is sensitive to model grid

resolution.

Fig 3 illustrates the evolution of the plume over the 8-day period in the C384L160 case ($\approx$25 km, 6 hPa resolution). The plume is initialized over Alaska, reaches eastern North America by Day 4, and Eurasia by Day 8, with strong filamentation along the way due to wind shear. Such rapid transport and filamentation is typical of free tropospheric plumes at northern mid-latitudes

(Stohl et al., 2002). The plume gradually subsides over the 8-day period, again typical of observations (Crawford et al., 2004). The spreading and dilution of the plume apparent in Fig. 3 is due in part to the plotting of column and meridional average VMRs for visualization purposes; the actual numerical diffusion is less and can be quantified by the VMR decay and entropy increase for the actual plume, as described below.

**4.2 Numerical diffusion at different grid resolutions**

Exact solution to the advection equation conserves the VMR, even for divergent or sheared flow (Chapter 7.2 of Brasseur and Jacob, 2017) . Our simulation includes advection as the only process. It follows that any VMR decay in the model plume must be due solely to numerical diffusion and provides a metric for this diffusion.

Fig. 4 shows the rate of decay of the maximum plume VMR for the different horizontal and vertical resolutions of our

simulations. The time scale for this decay diagnoses the rate of plume dissipation from numerical diffusion and can be used to compare different grid resolutions (Eastham and Jacob, 2017; Rastigejev et al., 2010).

At the lowest resolution (C48L20), the maximum VMR in the plume drops from 1.0 to 0.1 after 8 days. Such rapid diffusion is consistent with the mid-latitudes results of Eastham and Jacob (2017) using GEOS-FP winds. Starting from C48L20, solely

increasing the vertical resolution has no benefit in reducing numerical diffusion (Fig. 4, top left panel). Solely increasing horizontal resolution has some benefit for the first 4 days of aging, but by day 5 the benefit is gone (Fig. 4, bottom left panel).



This is consistent with the theory in Section 2.1 that inadequate resolution in one direction will limit the overall accuracy, making grid refinement in the other direction useless.

However, once the resolution of one dimension is high enough that it is no longer a limiting factor, grid refinement in the other direction becomes effective. This is illustrated in the right panels of Fig. 4. Increasing vertical resolution in a C384 simulation has sustained benefit from L20 to L160, and increasing horizontal resolution in a L160 simulation has sustained benefit from C48 to C384. At the highest resolution (C384L160), the decay in the maximum VMR is only 35% after 8 days of transport, a drastic improvement over the simulation cases presented by Rastigejev et al. (2010) and Eastham and Jacob (2017).

The behavior of decay rates in Fig. 4 lends further insights into numerical diffusion. We see that the decay rates are initially slow and then abruptly increase. This is because the plume is initially well resolved on the grid, but as the plume gradually filaments and becomes poorly resolved fast numerical diffusion takes over. Increasing horizontal resolution delays the onset of this fast numerical diffusion, as seen most dramatically in the bottom left panel of Fig. 4. Thus a factor in the choice of resolution should be the extent of time over which the model plumes must be preserved, considering that molecular diffusion will eventually dissipate the plumes in the actual atmosphere as they filament down to the millimeter Kolmogorov scale (Chapter 8 of Brasseur and Jacob 2017). Observations show that intercontinental free tropospheric plumes can retain their structure for at least a week (Heald et al., 2003; Zhang et al., 2008), so there is benefit in the highest range of resolutions investigated in our simulations.

Fig. 5 shows the vertical profile of maximum VMRs for each model level after 8 days of simulation, at the lowest model resolution (C48L20), the highest model resolution (C384L160), and intermediate cases where only horizontal or vertical resolution is increased from the low-resolution case (C384L20, C48L160). Starting from C48L20, solely increasing either the horizontal resolution (to C384L20) or the vertical resolution (to C48L160) has limited improvement on the vertical profile. This is the familiar picture of models being unable to preserve the vertical structure of pollution plumes on intercontinental scales (Heald et al., 2003). Increasing both horizontal and vertical resolutions (to C384L160) drastically improves the preservation of the vertical profile and largely fixes the problem. The surface concentrations are close to zero in all cases but this is because the FV3 dynamical core does not include boundary layer physics. From the concentrations at 900-950 hPa we can conclude that the high-resolution simulation when implemented in a full GCM would lead to much stronger localized impact of the subsiding plume on surface concentrations.

Maximum VMR in the plume is an extreme value diagnostic that is relevant for plume observation and impact but is an imperfect measure of plume dissipation (Eastham and Jacob, 2017). As shown in Fig. 3, the plume shears into multiple filaments as it ages but the maximum VMR diagnoses just one of these filaments. Also, numerical diffusion will first erode





the plume as its edges while preserving the maximum VMR at the center. Eastham and Jacob (2017) used the expanding size of the plume as an alternate diagnostic but this relies on an arbitrary concentration threshold.

As a more general diagnostic of plume preservation, we calculate the entropy that takes into account all grid cells in the global
domain (Lauritzen and Thuburn 2012). The entropy of a 3D VMR field can be calculated by

$$S = -k \sum_{i=1}^{n} m_i C_i \log C_i ,\tag{29}$$

where $n$ is the total number of grid cells of index $i$, $C_i$ is the VMR, $m_i$ is the mass of air in the grid cell, and $k$ is a scaling factor such that the initial entropy is unity. Pure advection conserves entropy but diffusion increases it, and $S$ is maximized when the VMR field $C$ becomes uniform (complete mixing). A non-monotonic advection scheme can unphysically decrease entropy,
but here we use strictly monotonic schemes in both horizontal and vertical, so the best possible simulation would preserve the entropy.

Fig. 6 shows the increase in entropy as the plume decays at different model grid resolutions. Results are similar to the maximum VMR diagnostic (Fig. 4) in showing the limiting effects of either horizontal or vertical resolution, and the benefit of coupling
the two to improve the simulation. One difference is the absence of a time lag for plume dissipation. Whereas the maximum VMR is initially sheltered from numerical diffusion if the plume is resolved by a number of grid cells, numerical diffusion erodes the plume edges and the thinner filaments and this is captured by the entropy diagnostic. The entropy diagnostic also shows a slowdown of plume dissipation with time, particularly at coarse resolution, and this is due to the smoothing of the plume that allows concentration gradients to be better represented by the numerical schemes. At that point, however, the plume
may already be too dissipated. Ultimately, the choice of maximum VMR or entropy as a diagnostic of plume dissipation may depend on the application, but the implied requirements for grid resolution are similar. This is discussed further below.

### 4.3 Optimal combination of horizontal and vertical grid resolution

The results from Section 4.2, following on the theoretical analysis of Section 2, show that preserving plumes in global models may be limited by either horizontal or vertical resolution. It follows that there must be an optimal ratio of horizontal to vertical
grid spacing $(\Delta x/\Delta z)_{opt}$ for simulating the global-scale transport of plumes, as there is for the dynamical criteria reviewed in the Introduction. We derived such a ratio from theoretical analysis in Section 2, and here we derive it from the FV3 plume simulations.

Fig. 7 illustrates the trade-offs between horizontal and vertical resolution in the FV3 plume simulations, presented in a similar
manner to the results of the theoretical analysis in Fig. 1. The contours measure the preservation of the plume after 8 days, as diagnosed by either the maximum VMR or the entropy, using the Day 8 data from Figs. 4 and 6 with additional simulations at intermediate resolutions to better define the contours. As in Section 2.3, we aim to maximize VMR and/or minimize entropy





under the computational trade-offs $\Delta z \Delta x^3 = P$ and $\Delta z \Delta x^2 = P$. The solid lines show the computational trade-offs. Along each trade-off line, it is generally beneficial to move away from $\Delta x/\Delta z < 100$ (the upper left region of Fig. 7, $\Delta x = 25$-$50$ km, $\Delta z = 0.6$ km) to $\Delta x/\Delta z \sim 1000$ (the bottom region of Fig. 7, $\Delta x = 50$-$200$ km, $\Delta z = 0.08$ km), since it leads to better preservation of the plume without incurring more computational cost. Thus we already see that the current generation of models ($\Delta x/\Delta z \sim 20$)

is out of balance in privileging horizontal over vertical resolution.

As in Section 2.3, the optimal ratio $(\Delta x/\Delta z)_{opt}$ is defined by the point where the computational trade-off line parallels the contour line. Different ratios $\Delta x/\Delta z$ are shown as yellow dashed lines in Fig. 7. For the $\Delta z \Delta x^2 = P$ trade-off (white solid lines), the optimal range of $\Delta x/\Delta z$ is in the range 700-1500, consistent with the theoretical derivation in Section 2.3 that $\Delta x/\Delta z \sim 1000$.

The $\Delta z \Delta x^3 = P$ trade-off (the red solid lines in Fig. 7) leads to a higher $(\Delta x/\Delta z)_{opt}$ around 1500, again consistent with the theoretical analysis.

We conducted sensitivity tests with plumes of different initial vertical thicknesses and found similar results. Thicker plumes have better initial preservation of the maximum VMR but this is rapidly dissipated as the plume filaments. Although the

theoretical analysis of Section 2 implies that $(\Delta x/\Delta z)_{opt}$ should depend on the plume size, this applies to the stretched rather than to the initial plume. During model transport, plumes of different initial thicknesses tend to be stretched to similar steady-state thicknesses where the stretching rate (thinning the plume) is balanced by the numerical diffusion rate (thickening the plume) (Rastigejev et al., 2010).

The estimated $(\Delta x/\Delta z)_{opt}$ should not greatly depend on the advection scheme used, since fast numerical diffusion occurs when the plume has filamented to the point where gradients cannot be resolved and any advection scheme collapses to first-order accurate. One concern is whether FV3 represents realistically the ratio of vertical-to-horizontal shear that would occur in a wet atmosphere; this should be tested in simulations in an actual GCM. Nevertheless, it appears that the vertical resolution requirements for global simulation of chemical plumes are much larger than the ratio $(\Delta x/\Delta z)_{opt} \sim 100$ derived from dynamical

concerns of resolving fronts and gravity waves, and that current models have woefully inadequate vertical resolutions.

## 5 Conclusions and implications for global modeling of chemical plumes

Current global models are unable to simulate the observed persistence of chemical plumes in the free troposphere on intercontinental scales. The plumes dissipate too rapidly due to numerical diffusion in sheared flow. This is a major problem for global simulations of atmospheric composition and for diagnosing intercontinental pollution influences on surface air

quality. We investigated how this problem could be solved through increasing horizontal and vertical grid resolutions, and in what optimal combination. We used for this purpose the GFDL-FV3 global dynamical core to perform plume transport simulations, driven by flow with realistic shear as generated from a baroclinic instability test. The flexibility of this dynamical



core allowed us to conduct simulations over cubed-sphere horizontal resolutions ranging from C48 ($\approx$200 km) to C384 ($\approx$25 km) and vertical resolutions ranging from L20 (50 hPa) to L160 (6 hPa).

We began with a theoretical analysis of the plume advection problem to show that numerical diffusion may be limited by either horizontal grid resolution ($\Delta x$) or vertical resolution ($\Delta z$). This analysis must take into account that increasing horizontal resolution is more costly than increasing vertical resolution, as expressed by $\Delta x^k \Delta z = P$ where $P$ denotes the amount of computational resources available and $k = 2$ (fixed time step) or $k = 3$ (time step adjusted for the CFL condition). We derived from this analysis an optimal ratio $(\Delta x/\Delta z)_{opt} \approx k(w/2u)(L/H)^2 \sim 1000$ for resolving the long-range transport of plumes, where $u$ and $w$ are the horizontal and vertical components of the wind, and $L$ and $H$ are the horizontal and vertical dimensions of the plume. This is much larger than the optimal ratio $(\Delta x/\Delta z)_{opt} \sim 100$ derived from dynamical considerations of resolving fronts and gravity waves. Current global atmospheric models have $\Delta x/\Delta z \sim 20$ in the free troposphere ($\Delta x \sim 10$ km, $\Delta z \sim 0.5$ km), with an emphasis on continued improvement in horizontal resolution to the neglect of vertical resolution. This explains why excessively fast dissipation of chemical plumes takes place in these models.

We applied the FV3 dynamical core to simulate the transport over 8 days of a chemically inert free tropospheric plume at northern mid-latitudes. Transport in the dynamical core is solely by advection, and exact solution should therefore preserve the initial volume mixing ratio (VMR) in the plume. We diagnosed numerical diffusion over the 8-day simulation by the decay of the maximum VMR and the increase in entropy. We demonstrated how improvements in preserving the plume during transport can be limited by either horizontal or vertical resolution, in a manner consistent with the theoretical analysis. Our highest-resolution simulation (C384L160) preserved the maximum VMR in the plume to within 35% after 8 days in strongly sheared flow, retained the vertical structure of the plume, and led to much larger local intercontinental impacts on surface air than the coarser-resolution simulations. The required vertical resolution in the free troposphere is 6 hPa ($\approx$ 80 m), considerably finer than in current global models.

There are strong reasons for GCMs to focus on horizontal resolution as computational resources increase, as this allows better representation of cyclogenesis and other aspects of the meteorological simulation. However, simulations of global chemical transport require higher vertical resolution in the free troposphere. Considering that the free troposphere accounts for only about a third of all vertical levels in the current generation of models, adding vertical resolution only to that part of the atmosphere would not be expensive. A proper vertical resolution in the free troposphere would also benefit the simulation of water vapor with implications for the radiative budget and for cloud formation. Within the framework of current GCMs, it may be possible to improve chemical transport by conducting off-line CTM simulations with high vertical resolution, interpolating the meteorological archive from the parent GCM. The feasibility of such hybrid-resolution simulations has been studied by Methven and Hoskins (1999).




# 6 Code availability

The FV3 source code was obtained from https://www.gfdl.noaa.gov/cubed-sphere-quickstart/. All scripts for model configuration and data analysis are available at https://github.com/JiaweiZhuang/FV3_util (Zhuang, 2017). A Python package named "cubedsphere" (https://github.com/JiaweiZhuang/cubedsphere, Zhuang and Rothenberg, 2017) was developed by the

lead author for analyzing data on the cubed-sphere grid. We use xarray (Hoyer and Hamman, 2017) to process NetCDF data that are larger than computer memory.

# Appendix A: Relating $\partial^2 C/\partial t^2$ and $\partial^2 C/\partial x^2$ through the advection equation

Following Odman (1997), we start with the 1-D advection equation

$$\frac{\partial C}{\partial t} + u\frac{\partial C}{\partial x} = 0. \tag{A1}$$

The equation is solved by the 1-D upwind scheme

$$\frac{C(n+1,i)-C(n,i)}{\Delta t} + u\frac{C(n,i)-C(n,i-1)}{\Delta x} = 0. \tag{A2}$$

Using the Modified Equation Approach introduced in Section 2, we find the numerical scheme better represents the equation

$$\frac{\partial C}{\partial t} + u\frac{\partial C}{\partial x} = -\frac{\Delta t}{2}\frac{\partial^2 C}{\partial t^2} + \frac{u\Delta x}{2}\frac{\partial^2 C}{\partial x^2} + o(\Delta t + \Delta x). \tag{A3}$$

Apply the operation $(\Delta t/2)(\partial/\partial t)$ to Eq. (A3),

$$\frac{\Delta t}{2}\frac{\partial}{\partial t}\left[\frac{\partial C}{\partial t} + u\frac{\partial C}{\partial x}\right] = \frac{\Delta t}{2}\frac{\partial}{\partial t}\left[-\frac{\Delta t}{2}\frac{\partial^2 C}{\partial t^2} + \frac{u\Delta x}{2}\frac{\partial^2 C}{\partial x^2} + o(\Delta t + \Delta x)\right]. \tag{A4}$$

The right hand-side of Eq. (A4) only contains high-order terms such as $\Delta t^2$ and $\Delta t\Delta x$ so can be simply written as $o(\Delta t + \Delta x)$:

$$\frac{\Delta t}{2}\frac{\partial^2 C}{\partial t^2} + \frac{u\Delta t}{2}\frac{\partial^2 C}{\partial t\partial x} = o(\Delta t + \Delta x). \tag{A5}$$

Perform Eq. (A3) – Eq. (A5) to cancel the $(\Delta t/2)(\partial^2 C/\partial t^2)$ term:

$$\frac{\partial C}{\partial t} + u\frac{\partial C}{\partial x} = \frac{u\Delta t}{2}\frac{\partial^2 C}{\partial t\partial x} + \frac{u\Delta x}{2}\frac{\partial^2 C}{\partial x^2} + o(\Delta t + \Delta x). \tag{A6}$$

Compared to Eq. (A3), the second-order time derivative $(-\Delta t/2)(\partial^2 C/\partial t^2)$ is now replaced by the mixed-derivative

$(u\Delta t/2)(\partial^2 C/\partial t\partial x)$.



To further eliminate this mixed-derivative, apply the operation $(u\Delta t/2)(\partial/\partial x)$ to Eq. (A6)

$$\frac{u\Delta t}{2}\frac{\partial}{\partial x}\left[\frac{\partial C}{\partial t}+u\frac{\partial C}{\partial x}\right]=\frac{u\Delta t}{2}\frac{\partial}{\partial x}\left[-\frac{\Delta t}{2}\frac{\partial^2 C}{\partial t^2}+\frac{u\Delta x}{2}\frac{\partial^2 C}{\partial x^2}+o(\Delta t+\Delta x)\right].$$

(A7)

Again, the right hand-side of Eq. (A7) can be simply written as $o(\Delta t + \Delta x)$:

$$\frac{u\Delta t}{2}\frac{\partial^2 C}{\partial x\partial t}+\frac{u^2\Delta t}{2}\frac{\partial^2 C}{\partial x^2}=o(\Delta t+\Delta x).$$

(A8)

Perform Eq. (A6) + Eq. (A8) to cancel the $(u\Delta t/2)(\partial^2 C/\partial x\partial t)$ term

$$\frac{\partial C}{\partial t}+u\frac{\partial C}{\partial x}=-\frac{u^2\Delta t}{2}\frac{\partial^2 C}{\partial x^2}+\frac{u\Delta x}{2}\frac{\partial^2 C}{\partial x^2}+o(\Delta t+\Delta x).$$

(A9)

Now all time derivatives except the original $\partial C/\partial t$ are removed. The time step $\Delta t$ can also be removed by introducing the CFL number $\alpha = u\Delta t/\Delta x$. The first term on the right-hand side of Eq. (A9) becomes

$$-\frac{u^2\Delta t}{2}\frac{\partial^2 C}{\partial x^2}=-\alpha\frac{u\Delta x}{2}\frac{\partial^2 C}{\partial x^2}.$$

(A10)

Thus Eq. (A9) can be further simplified to

$$\frac{\partial C}{\partial t}+u\frac{\partial C}{\partial x}=(1-\alpha)\frac{u\Delta x}{2}\frac{\partial^2 C}{\partial x^2}+o(\Delta t+\Delta x).$$

(A11)

From Eq. (A3) to Eq. (A11), the time-derivative $(\Delta t/2)(\partial^2 C/\partial t^2)$ is approximated by the spatial derivative $(\alpha u\Delta x/2)(\partial^2 C/\partial x^2)$. This means that, as long as the CFL condition is satisfied, the time discretization error will not limit the overall accuracy. This conclusion still applies to a 3-D advection equation, although the above mathematical derivation will produce mixed

derivatives like $\partial^2 C/\partial x\partial y$, so a compact formula like Eq. (A11) cannot be easily obtained.

**Appendix B: Comparing vertical numerical diffusion in FV3 and TPCORE schemes**

Here we use the GEOS-Chem CTM to compare vertical numerical diffusion in FV3's advection scheme to that in TPCORE, a 3D Eulerian advection scheme (Lin and Rood, 1996). TPCORE is the standard advection scheme in the "classic" version of the GEOS-Chem CTM (Bey et al., 2001), while FV3 is used in the High-Performance version of GEOS-Chem (GCHP; Long

et al., 2017; Yu et al., 2017). Unlike FV3, TPCORE uses a regular latitude-longitude geometry and a vertically-Eulerian discretization. When the CFL number is less than one, the horizontal tracer transport uses the monotonic PPM as in FV3;



otherwise, a semi-Lagrangian method is used. The vertical tracer transport uses PPM with Huynh's second constraint (Huynh, 1997). We use a C48 horizontal resolution for GEOS-Chem with FV3 and a corresponding 2°×2.5° resolution for GEOS-Chem with TPCORE. Both versions use the native GEOS-FP 72-level hybrid sigma-pressure vertical coordinate and a time step of 15 minutes.

We use the idealized Hadley-like circulation test in the 2012 Dynamical Core Model Intercomparison Project (Kent et al. 2014) to benchmark the vertical diffusion in both models. The simulation is illustrated by Fig. B1. The initial tracer layer (Fig. B1, left panel) is advected in the vertical by a Hadley-like flow (Fig. B1, middle panels) and then gets reverted to the original state by a reverse flow (Fig. B1, right panels). The true solution at the final state should be the same as the initial condition, and the

deviation from the initial condition is due to numerical error. The error norm can be calculated by

$$l = \frac{\sum_{i=1}^{n} m_i |C_i - C_{i,true}|}{\sum_{i=1}^{n} m_i |C_{i,true}|} \qquad (B1)$$

where $n$ is the total number of grid cells of index $i$, $m_i$ is the mass of air in the grid cell, $C_i$ is the VMR at the final state and $C_{i,true}$ is the VMR at the initial state. We find $l = 19.0\%$ for TPCORE and $l = 16.2\%$ for FV3, indicating that the vertically

Lagrangian scheme in FV3 has a diffusion similar to the Eulerian scheme in TPCORE.

*Acknowledgements.* The authors would like to thank Lucas Harris, Xi Chen and S. J. Lin at GFDL for general guidance on the GFDL-FV3 model. Resources supporting this work were provided by the NASA High-End Computing (HEC) Program

through the NASA Advanced Supercomputing (NAS) Division at Ames Research Center. This work was supported by the NASA Modeling, Analysis, and Prediction (MAP) Program.

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

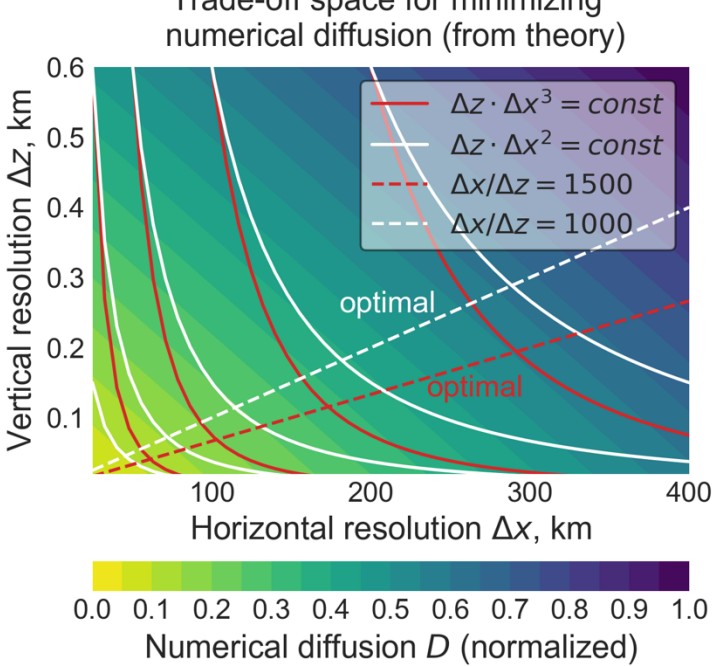

**Figure 1.** Optimal combination of horizontal and vertical grid resolutions ($\Delta x$ and $\Delta z$) for minimizing numerical diffusion of chemical plumes within a given amount of computational resources. Results are from the theoretical analysis of Section 2.3. The filled contours show the magnitude of the numerical diffusion term $D$ from Eq. (23) as a function of $\Delta x$ and $\Delta z$ with $B/A = 500$. The solid lines indicate a fixed amount of computational resources ($P$) in the trade-off between horizontal and vertical resolution, either using a fixed time step ($\Delta z \Delta x^2 = P$) or accounting for the CFL condition ($\Delta z \Delta x^3 = P$). The dashed lines from Eq. (27) indicate the corresponding optimal $\Delta x / \Delta z$ ratios to minimize numerical diffusion.





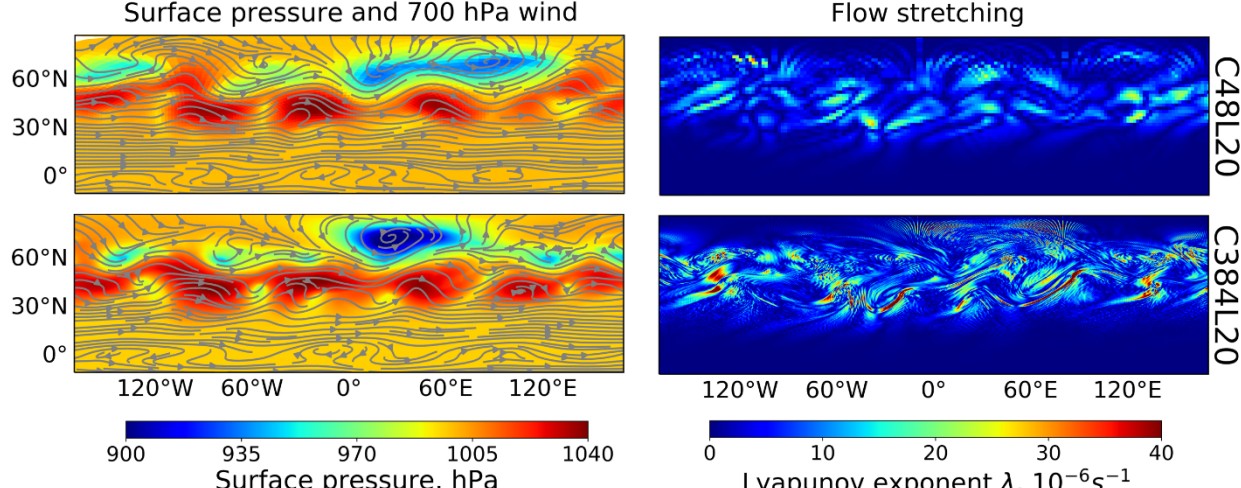

**Figure 2.** Atmospheric flow generated by the FV3 dynamical core in a baroclinic instability test, 16 days after initialization of the test and 8 days after the release of the chemical plume. Shown are surface pressures, 700 hPa flow streamlines, and Lyapunov exponents $\lambda = \partial u/\partial x$ measuring the stretching of the flow. The top row shows results from the lowest horizontal resolution (C48, ≈200 km) and the bottom row shows results from the highest horizontal resolution (C384, ≈25 km), both with 20 vertical levels (L20). Increasing vertical resolution has little effect on the dynamics, as discussed in the text.

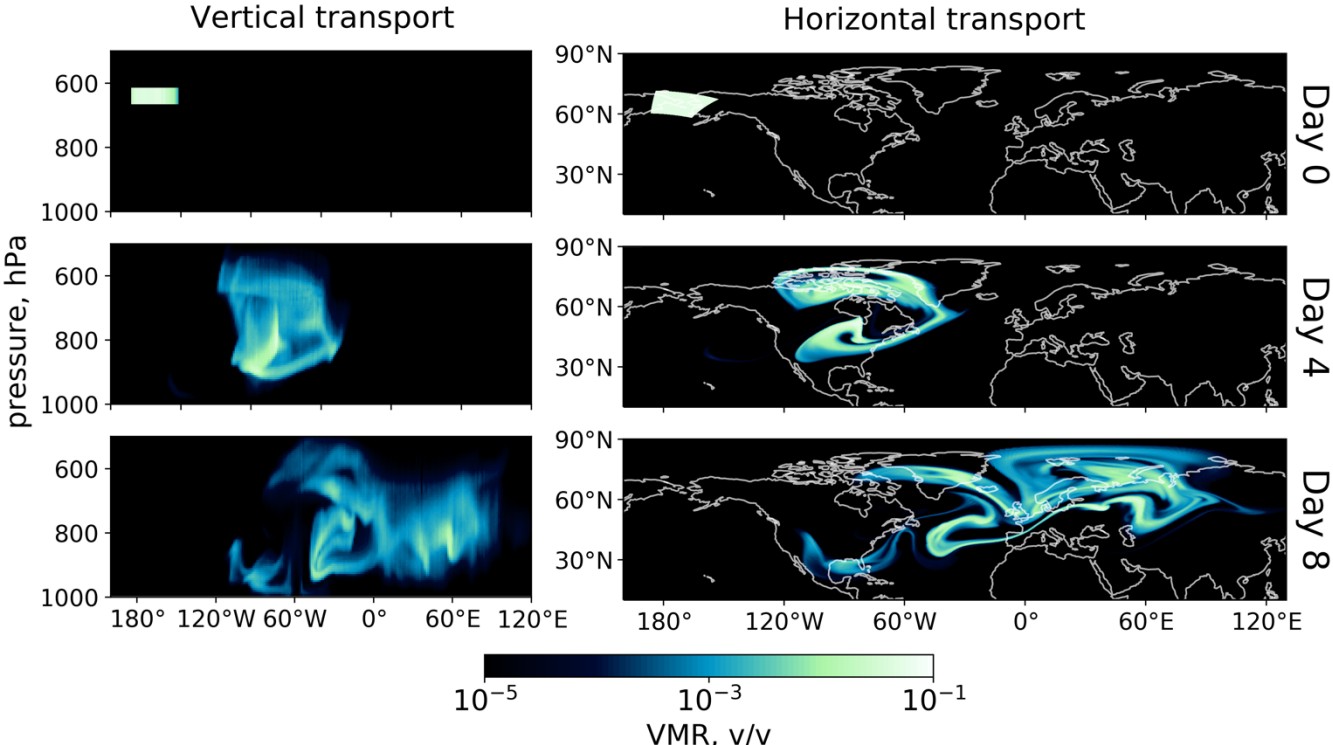

**Figure 3.** 8-day simulation of plume transport in the FV3 dynamical core at C384L160 resolution (≈25 km in horizontal and 6 hPa in vertical). The plume is initialized at 625 hPa over Alaska with a volume mixing ratio (VMR) of unity over a domain 1000×1000 km$^2$ in the horizontal and 50 hPa thickness in the vertical. The left panels show the vertical and longitudinal transport of the plume as the meridionally averaged VMR. The right panels show the horizontal transport of the plume as column-averaged mixing ratios. Because VMRs are plotted here as meridional or vertical averages, the values are much lower than the actual values in the plume.





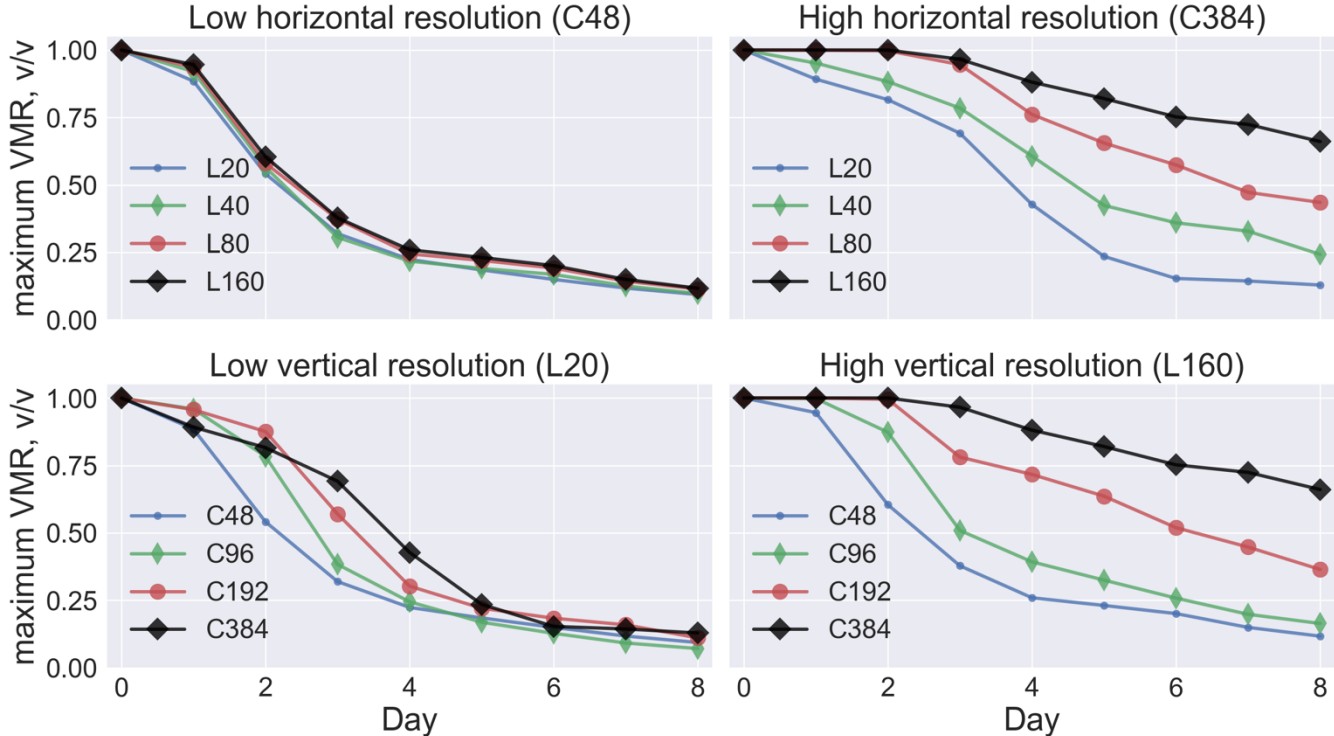

**Figure 4.** Plume decay due to numerical diffusion at different model grid resolutions. The plume is released in the free troposphere at northern mid-latitudes with an initial volume mixing ratio (VMR) of unity. Plume decay is measured by the decrease in the maximum VMR as a function of time. Model horizontal resolution is defined by a cubed-sphere grid ranging from C48 (≈200 km) to C384 (≈25 km). Vertical resolution is defined by an equally spaced isobaric grid ranging from L20 (20 levels, each 50 hPa thick) to L160 (160 levels, each 6 hPa thick).

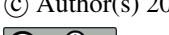


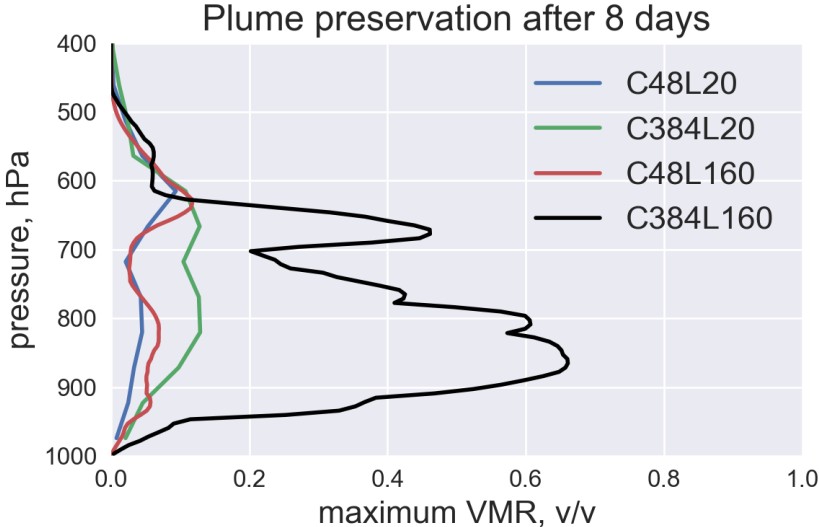

**Figure 5.** Vertical profile of maximum VMR for each model vertical level after 8 days of simulation, at low model resolution (C48L20), high model resolution (C384L160), and intermediate cases where only horizontal resolution or vertical resolution is increased from the low-resolution case (C384L20, C48L160).




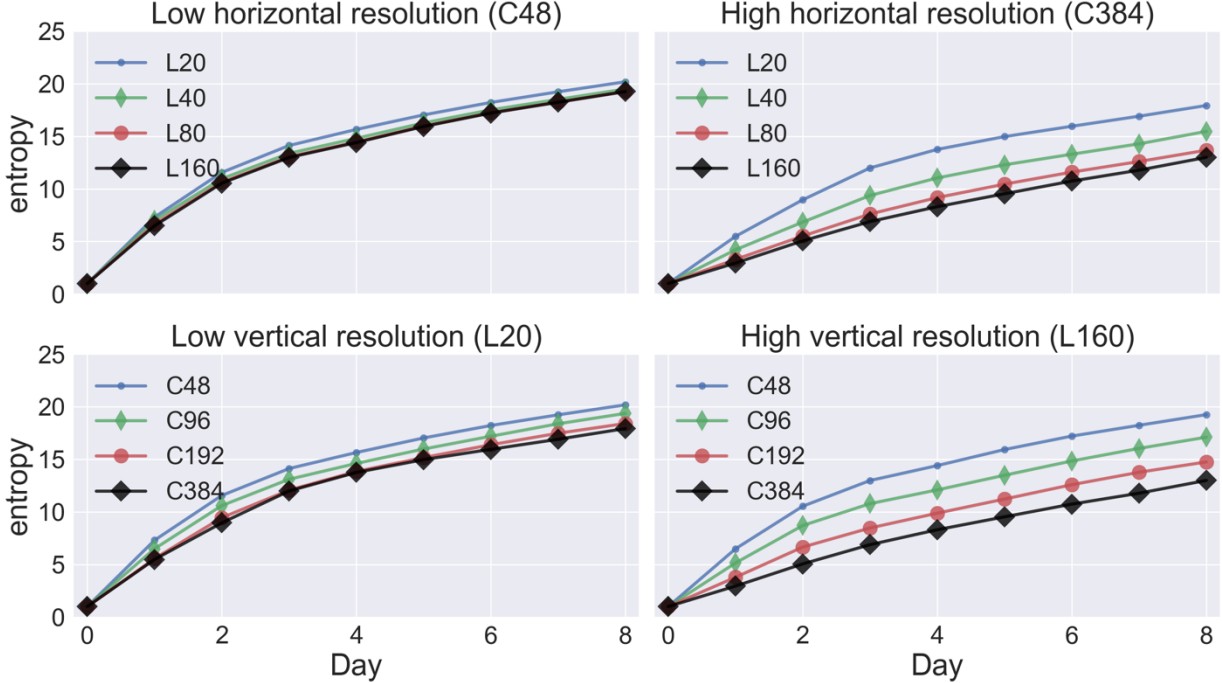

**Figure 6.** Same as Fig. 4 but with entropy instead of maximum VMR as a diagnostic for numerical diffusion. The entropy is initialized on Day 0 with a value of 1. Pure advection conserves entropy but diffusion increases it.




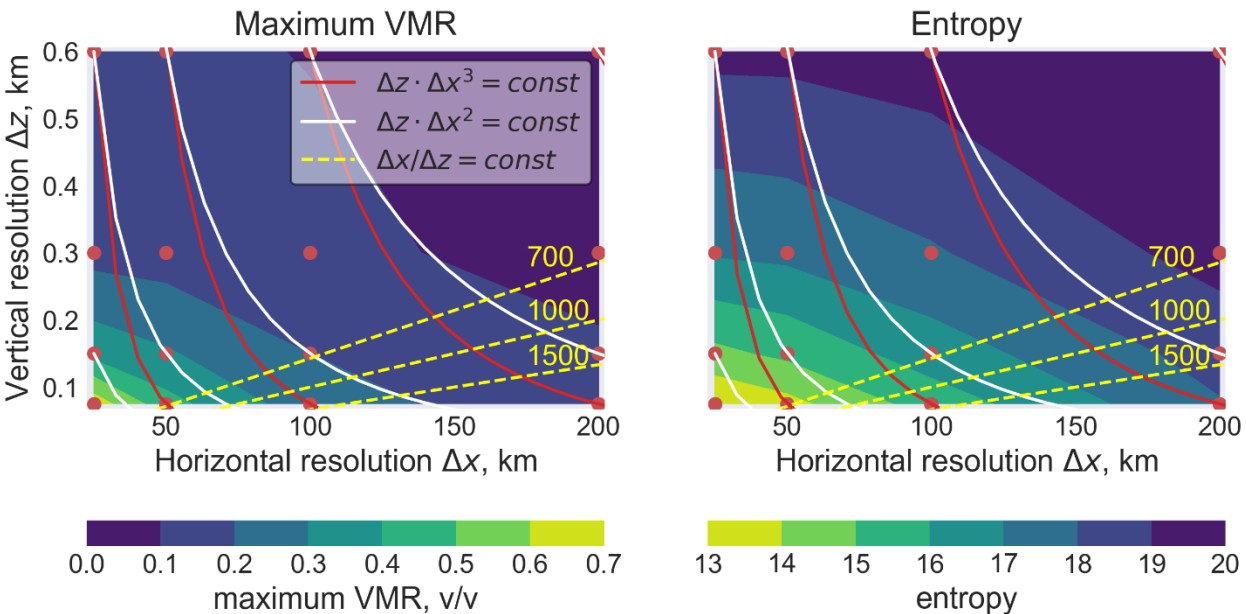

**Figure 7.** Optimal combination of horizontal and vertical grid resolutions ($\Delta x$ and $\Delta z$) for minimizing numerical diffusion of chemical plumes within a given amount of computational resources. Results are from the FV3 simulation (data from Fig. 4 and 6) and can be compared to Fig. 1 that shows similar results from theoretical analysis. The filled contours show the maximum plume VMR (left panel) or entropy (right panel) on Day 8 of the simulations as metrics of numerical diffusion. High maximum VMR and low entropy are indicative of low numerical diffusion. The red dots are the data points used to construct the contours, each point corresponding to a simulation at a given resolution. The straight lines indicate a fixed amount of computational resources ($P$) in the trade-off between horizontal and vertical resolution, either using a fixed time step ($\Delta z \Delta x^2 = P$) or accounting for the CFL condition ($\Delta z \Delta x^3 = P$). The yellow dashed lines show different $\Delta x / \Delta z$ ratios (700, 1000, 1500).

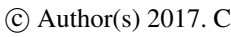



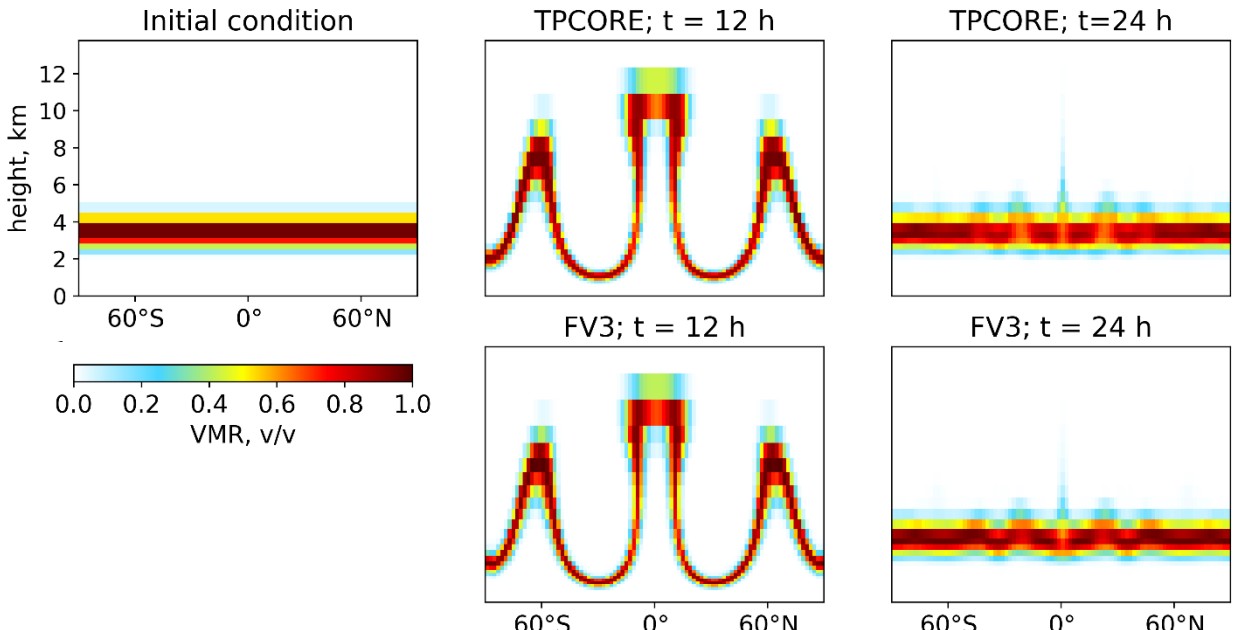

**Figure B1.** Comparing vertical diffusion in the GEOS-Chem CTM using either the TPCORE Eulerian advection scheme (top panels) or the FV3 vertically Lagrangian advection scheme (bottom panels). A Hadley-like circulation test is applied to both schemes with rising motion in the first 12 hours followed by return to the original state in the next 12 hours (Kent et al., 2014). The tracer field is independent of longitude. The true solution at the final state ($t = 24$ h) should be the same as the initial condition, and the deviation from the initial condition is due to numerical error.