# Peer review of "The importance of vertical resolution in the free troposphere for modeling intercontinental plumes"

_Atmospheric Chemistry and Physics, 2017_

## Referee Comment (RC1) · Anonymous Referee #1 · 7 Jan 2018

Report on 'The importance of vertical resolution in the free troposphere for modeling intercontinental plumes' by Zhuang et al.

This paper presents a discussion of numerical diffusion in solving tracer transport problems with some theoretical analysis and numerical calculations. It is argued that vertical resolution is, in current models, often more of a problem than horizontal resolution in avoiding excessive diffusion and plume dilution. Increasing vertical resolution should be given higher priority than horizontal resolution when trying to optimise the results achievable with a given computational resource.

General comments:

The topic is of importance, the paper is well written and well argued, and the simulation results appear credible and useful. However there are a number of aspects where there are alternative perspectives and it would be useful if the paper could reflect some of these. I have described these 'perspective issues' below. The authors may or may not agree with these issues. This is fine, but, either way, there would be benefit in discussing the issues and/or in providing further arguments for their own view point which might convince readers with a different view point. I also have some specific comments which are not to do with perspective. In terms of clarity and presentation, the paper is excellent and I have suggested very few technical corrections.

Comments on perspective:

Suppose we have a specified velocity field and we are trying to compute the tracer field. Over time the tracer plume will be stretched into thin filaments. Imagine that one has a high resolution simulation which resolves all the features in the tracer at a given time. The best low resolution description could (arguably) then be obtained by averaging the high resolution field to the lower resolution. This low resolution description will not preserve the peak volume mixing ratio. Hence one could argue that seeking to maximise VMR or minimise numerical diffusion is not really the right thing to do; one should seek the right amount of diffusion for the resolution used. Page 12, lines 11-16 go some way to acknowledge this situation but it could be reflected better in other parts of the paper. E.g. page 13, line 10 'the best possible simulation would preserve the entropy'. In practice maximising VMR or minimising diffusion or entropy gain are likely to be OK because it's hard to have too little diffusion without numerical schemes which generate unsatisfactory solutions (loss of monotonicity, negative concentrations), but a little more discussion would be useful (and also see next comment).

In real flows (with molecular diffusion, no matter how small) the filamentation will cascade down to very small scales and eventually be smoothed out by diffusion. Hence, although it's correct to say advection preserves VMR, this doesn't apply to real flows without qualification and hence the relevance of the statement could benefit from some discussion (see page 4, lines 9-11 and page 11, lines 20-22). Also if one allows resolution of small scales in the tracer but doesn't add in smaller scales in the driving flow field, this is like the viscous-convective $k^{-1}$ spectral range in turbulent flows when the molecular diffusivity is much less than the viscosity (see e.g. Monin and Yaglom, Statistical Fluid Mechanics, vol 2, p436, where the tracer is temperature). This can lead to less diffusion

than would occur if the smaller velocity scales were present and to rougher tracer fields. ($k^{-1}$ decays slower with increasing wave number than the standard $k^{-5/3}$ inertial sub-range). So perhaps this is a situation where one could have too little diffusion (see page 15, lines 30-32) and too much small scale structure in the tracer field.

Mathematically it's nice to think of having a well defined tracer problem (an advection or advection-diffusion problem with a specified flow field) which one tries to solve more and more accurately as computing power increases. This seems to be the authors perspective in places. However it's probably more useful to change the problem by resolving more of the driving flow field. The comment immediately above reflects this to some extent, but one can also resolve qualitatively different features such as convective updrafts and, eventually, boundary layer turbulence. Here the physics is different, vertical velocities may be larger, and a more isotropic grid may be appropriate. This is relevant to the various discussions of $\Delta x/\Delta z$; for example on page 8, $A$ and $B$ may be similar in size at high resolution. On page 11, line 6, 'offsets the gains' seems to imply that the authors think the increased resolution of small scale eddies is a backwards step (presumably judged by the narrow target of minimising diffusion). However the simulation may well be better overall, and the material can't diffuse far because these new eddies are small, so the large scale distribution of the tracer shouldn't be affected much.

Other specific comments:

Page 1, line 18: I'm not sure 'dissipate' is the right word. 'Disperse' might be better. The former suggests loss of mass rather than spreading out and dilution. Of course mass loss is not impossible if mass conservation isn't satisfied but I'm not sure that interpretation is intended here. See also page 12, line 32, page 13, line 20, page 14, line 28 and figure 4 caption ('plume decay'). I have a similar comment with 'Preserve the plume' (page 10, line 17) and 'subsides' (page 11, line 15). The material is preserved but it's dispersed and diluted, and presumably the authors don't mean subside in the sense of moving down towards the ground. (I guess they might actually mean subside as the plume does spread downwards. It's slightly curious that nothing mixes up to higher heights within the troposphere. But this is likely to reflect this case only rather than generic behaviour.)

Page 3, lines 13-18: I think '(native)' means the resolution of the driving flow. If so then it would be good to say what was done at lower resolutions – hopefully the driving flow was smoothed rather than applying one grid point value over a larger area where it may be unrepresentative. Also I'm not sure what a '2-D horizontal plume' is in this context. Does this mean concentration is independent of height and depends only on $x$ and $y$?

Eqns 19-21: While this is indicative, it's based on the idea of a coherent smoothly varying plume. In reality, once the plume is broken up into filaments with a lot of small scale structure, the second order derivatives will be dominated by the small scales, not the large scales which are characterised by $L$ and $H$.

Page 8: This argument would not work if one was considering 3-D isotropic turbulence. Here one would expect $k = 2$ (or at least the power of $\Delta x$ would be twice the power of $\Delta z$ – both would have a CFL related increase), but it doesn't make sense to make $\Delta z$ smaller

(or bigger) than $\Delta x$. Why does the argument go wrong for this case? One explanation for this is that perhaps one should replace (23) with $D = \max(A\Delta x, B\Delta z)$, consistent with page 8 lines 29-30, or replace it with $D = A\Delta x + A\Delta y + B\Delta z$, treating $x, y, z$ on an equal footing. I think the authors either need to use one of these more balanced forms of $D$ (with minor changes in the conclusions) or provide more justification for their choice.

Page 10, line 27 to page 11, line 3: It might be better to initially leave the description of $\lambda$ in words and not introduce $\partial u / \partial x$ (see fig 2 caption too) only to redefine it in (28). More importantly, stretching is not in general aligned with the local flow. One can add a uniform flow (Galilean transformation) in any direction without altering the stretching properties of the flow. The values are only used qualitatively however, so it's probably OK to use this as an indication of stretching.

Technical corrections:

Page 9, line 27: Should 'initialize' be 'run' or 'calculate'?

Page 11, line 8: 'Large scale vertical wind speeds' might be better. The values are too small for convective updrafts or boundary layer turbulence.

Fig 7 caption: 'straight lines' should be 'solid lines'.

---

## Referee Comment (RC2) · Anonymous Referee #2 · 9 Jan 2018

Manuscript ID acp-2017-1124 entitled 'The importance of vertical resolution in the free troposphere for modelling intercontinental plumes'.

The aim of this paper is to determine the horizontal and vertical resolution required for accurate free tropospheric plume transport. This topic is of interest and the analysis in this paper is presented clearly and precisely. The quality of the writing is excellent and there are therefore few specific comments. I recommend the paper for publication in ACPD with minor corrections.

General Comments

1. The paper deals specifically with free tropospheric tracer transport and situations in which the vertical velocities are small (order 1cm $s^{-1}$). Do the results hold for different situations in which vertical velocities are larger? For example, in a convective boundary layer, A and B (eq 24) may be similar so B/A is much smaller than 500.

2. In the abstract the authors claim that 'the local surface pollution influence from the subsiding plume on intercontinental scales is also considerably increased', but on p12 say that this result is just implied, as the fV3 dynamical core does not include boundary layer physics. Can the authors justify the term 'considerably' increased as they have no quantitative estimate of this potential increase?

3. Throughout the paper the authors refer to typical horizontal plume scales of 1000km. Given that in the real atmosphere plumes sizes cover a whole spectrum how sensitive are the results to the initial plume size? The authors present sensitivity results for plumes of different initial vertical thicknesses but not different initial horizontal spreads. On a similar topic of variable resolution, I was surprised that there was no discussion of local grid refinement, grid transformation or irregular grids as alternative methods to gain numerical efficiency in the introduction.

Specific Comments
1. Page 2 line 6: the authors use 'thick plumes of Asian ozone pollution', do they refer to optical or geometric thickness here?
2. Page 4 line 14: This sentence is a little difficult to follow. What is 'positive'?
3. Page 5 line 21: CFL acronym should be defined.
4. Page 6 line 5: 'This explains why', what is 'this'? Are the authors referring to equation 14?
5. Page 7 line 14 and elsewhere: The authors often refer to 'reviewed above', 'will be discussed', 'described below' etc. Please can the specific section numbers be used in these places?
6. Page 13 line 4: 'S' is not defined.
7. Page 14 line 6: Why is the z underlined here?
8. Page 30 figure B1: A difference plot might help to highlight deviations from the initial conditions due to numerical error more clearly.

---

## Referee Comment (RC3) · Anonymous Referee #3 · 12 Jan 2018

Review of "The importance of vertical resolution in the free troposphere for modelling intercontinental plumes" by Zhuang et al.

This paper discusses the importance of vertical resolution relative to the horizontal resolution. The authors claim that there exists an optimal grid resolution ratio of 1000 between horizontal and vertical resolution.

The statement that the often used approach of increasing the horizontal resolution alone is insufficient for an optimal solution has been made before, also by the same authors. Also, as the authors themselves point out, optimal grid resolution ratios not too dissimilar to what is stated in the manuscript have been suggested before. This study is extension to previous work in that it aims to theoretically derive an optimal grid resolution ratio and to support the findings by performing idealised simulations with the GFDL-FV3 global dynamical core. Though not ground breaking the manuscript could be published after revisions as outlined below.

The paper makes very bold statements that are not widely valid and applicable since the findings are much more model, case and resolution dependent. This needs to be acknowledged and discussed more widely.

For increasingly higher spatial resolutions that approach convection permitting or even convection resolving scales, an aspect ratio of 1000 wouldn't make much sense since the resolved dynamical features become more isotropic compared to a model where most of the vertical motions is on the subgrid scale.

For a plume with an aspect ratio of 1000 (and flow ratios in the same order) as used here it is not surprising that a grid with the same aspect ratio provides an optimal solution.

A validation with a real event would be limited by an incomplete knowledge of the atmospheric state. Only with a perfect knowledge of the atmospheric flow that drives the tracer transport the presented argument holds.

Advection is not the only time step limiting process. Gravity waves, for example, pose a much stronger constrain. Since in global models the vertical resolution is higher than the vertical resolution (even if smaller than an aspect ratio of 1000) the numerical treatment is usually not the same for the vertical and the horizontal. GFDL-FV3 is an extreme example since it is not even Eulerian in the vertical. Although this is mentioned in section 3 this should already be mentioned at the beginning of section 2 since GFDL-FV3 is already discussed in section 1. Even then GFDL-FV3 doesn't fit the description of an Eulerian chemical transport model for which the theoretical analysis is performed in section 2. GFDL-FV3 is a full dynamical core with tracer transport so that the analysis in section 2 only applies to this model if aspects of predicting the flow are ignored. However, a more accurate prediction of the flow with increasing horizontal and/or vertical resolution is key for the interpretation of results in section 4. Appendix B is important but it shows similar numerical diffusion properties only for the transport, not for the dynamical core as a whole if I understand correctly.

Volume mixing ratio is usually defined as tracer mass per unit volume and is thus not preserved in a non-divergent flow as stated. However, mass mixing ratio (the tracer mass per total mass) is. According to the figure labels, the authors define volume mixing ratio as tracer volume per total volume. This is only equivalent to mass mixing ratio for gaseous tracers, not for particulate tracers, like aerosols. To my knowledge, GFDL-FV3 solves for the tracer mass per unit dry air.

I would have expected that section 4 confirms the optimal grid resolution ratio of 1000 but it does not. Since section 4 only focuses on error metrics for the tracer transport without including computer time, unsurprisingly, the case with the highest horizontal and vertical resolution produces the best results. At the highest resolution the aspect ratio is below 1000 (dx/dz=25km/80m=312.5) so that simulations with higher vertical resolution would be needed to be able to prove the point of section 2. Only for C48 do the performed simulations significantly exceed an aspect ratio of 1000. Simulations with higher vertical resolution at higher horizontal resolution are needed to prove a relationship based on equation (25).

The initial conditions chosen for the test case are discontinuous at the plume edges. No matter what the spatial resolution, the grid well never resolve the transition between inside and outside plume. Thus, this test case cannot converge. It should still be possible to test the relationship for an optimal grid ratio as derived in section 2 but it would have been better to consider a test case that can numerically converge.

---

## Author Comment (AC1) · 19 Feb 2018

Dear Editor,

Thank you for considering our submission and for arranging the careful reviews. Our revised manuscript is enclosed; please find below our responses (in blue) to each of the comments made by the reviewers.

**Reviewer #1**

This paper presents a discussion of numerical diffusion in solving tracer transport problems with some theoretical analysis and numerical calculations. It is argued that vertical resolution is, in current models, often more of a problem than horizontal resolution in avoiding excessive diffusion and plume dilution. Increasing vertical resolution should be given higher priority than horizontal resolution when trying to optimise the results achievable with a given computational resource.

General comments:

The topic is of importance, the paper is well written and well argued, and the simulation results appear credible and useful. However there are a number of aspects where there are alternative perspectives and it would be useful if the paper could reflect some of these. I have described these 'perspective issues' below. The authors may or may not agree with these issues. This is fine, but, either way, there would be benefit in discussing the issues and/or in providing further arguments for their own view point which might convince readers with a different view point. I also have some specific comments which are not to do with perspective. In terms of clarity and presentation, the paper is excellent and I have suggested very few technical corrections.

Comments on perspective:

Suppose we have a specified velocity field and we are trying to compute the tracer field. Over time the tracer plume will be stretched into thin filaments. Imagine that one has a high resolution simulation which resolves all the features in the tracer at a given time. The best low resolution description could (arguably) then be obtained by averaging the high resolution field to the lower resolution. This low resolution description will not preserve the peak volume mixing ratio. Hence one could argue that seeking to maximise VMR or minimise numerical diffusion is not really the right thing to do; one should seek the right amount of diffusion for the resolution used. Page 12, lines 11-16 go some way to acknowledge this situation but it could be reflected better in other parts of the paper. E.g. page 13, line 10 'the best possible simulation would preserve the entropy'. In practice maximising VMR or minimising diffusion or entropy gain are likely to be OK because it's hard to have too little diffusion without numerical schemes which generate unsatisfactory solutions (loss of monotonicity, negative concentrations), but a little more discussion would be useful (and also see next comment).

We agree that there could exist an "representation error", which is sorely because the low-resolution grid cannot represent the peak VMR due to spatial averaging. We have examined this type of error in Fig. C1 (see below). The tracer field regridded from C384 to C48 is significantly less diffusive than the tracer field produced directly by a C48 simulation. Thus, the C48 grid is actually able to represent relatively sharp tracer gradient. The C48 simulation fails to preserve the sharp gradient because the numerical error in solving the advection equation is too large, not because the C48 grid itself is unable to resolve such gradient.

[Figure]

Fig C1. Column-averaged VMR at Day 8 of the simulation. The highest resolution result (C384L160, top panel) is regridded to lower horizontal resolution (C48, middle panel). The regridded result is still significantly less diffusive than a C48L160 simulation (bottom panel).

In real flows (with molecular diffusion, no matter how small) the filamentation will cascade down to very small scales and eventually be smoothed out by diffusion. Hence, although it's correct to say advection preserves VMR, this doesn't apply to real flows without qualification and hence the relevance of the statement could benefit from some discussion (see page 4, lines 9-11 and page 11, lines 20-22). Also if one allows resolution of small scales in the tracer but doesn't add in smaller scales in the driving flow field, this is like the viscous-convective $k-1$ spectral range in turbulent flows when the molecular diffusivity is much less than the viscosity (see e.g. Monin and Yaglom, Statistical Fluid Mechanics, vol 2, p436, where the tracer is temperature). This can lead to less diffusion than would occur if the smaller velocity scales were present and to rougher tracer fields. ($k-1$ decays slower with increasing wave number than the standard $k-5/3$ inertial sub- range). So perhaps this is a situation where one could have too little diffusion (see page 15, lines 30-32) and too much small scale structure in the tracer field.

Thank you for bringing up this issue. We have added a paragraph to the introduction explaining that a model with no advection error would indeed underestimate diffusion. We cite D'Isidoro et al., (2010) who compared the magnitudes of numerical and actual (subgrid) turbulent diffusion, They find that numerical diffusion dominates on the scales of interest here, and indeed this is seen in the inability of the models to preserve plumes.

Mathematically it's nice to think of having a well defined tracer problem (an advection or advection-diffusion problem with a specified flow field) which one tries to solve more and more accurately as computing power increases. This seems to be the authors perspective in places. However it's probably more useful to change the problem by resolving more of the driving flow field.

We can actually better resolve the tracer field without resolving more of the driving flow field (Methven and Hoskins, 1999). Even if the driving flow field is very smooth and can be resolved by a very low-resolution grid, the tracer field can have large gradients and thus needs higher resolution. An extreme example is 1-D advection under constant wind – the wind field can be resolved by a single grid cell, but resolving the tracer field would require many cells.

The comment immediately above reflects this to some extent, but one can also resolve qualitatively different features such as convective updrafts and, eventually, boundary layer turbulence. Here the physics is different, vertical velocities may be larger, and a more isotropic grid may be appropriate. This is relevant to the various discussions of $\Delta x/\Delta z$; for example on page 8, A and B may be similar in size at high resolution.

Our focus is on the free troposphere under stable conditions, hence convective updrafts and boundary layer turbulence are not relevant. This is now clarified in the abstract and in section 4.1, where we also recognize that the plume may originate from a convective updraft, and may dissipate in the boundary layer, but we do not aim to simulate these processes.

On page 11, line 6, 'offsets the gains' seems to imply that the authors think the increased resolution of small scale eddies is a backwards step (presumably judged by the narrow target of minimising diffusion). However the simulation may well be better overall, and the material can't diffuse far because these new eddies are small, so the large scale distribution of the tracer shouldn't be affected much.

We have rephrased "offsets the gains".

Other specific comments:

Page 1, line 18: I'm not sure 'dissipate' is the right word. 'Disperse' might be better. The former suggests loss of mass rather than spreading out and dilution. Of course mass loss is not impossible if mass conservation isn't satisfied but I'm not sure that interpretation is intended here. See also page 12, line 32, page 13, line 20, page 14, line 28 and figure 4 caption ('plume decay').

We have replaced "dissipation" and "decay" by "dilution". "Dispersion" would not be accurate because it refers to a phase error in the numerical solution.

I have a similar comment with 'Preserve the plume' (page 10, line 17) and 'subsides' (page 11, line 15). The material is preserved but it's dispersed and diluted, and presumably the authors don't mean subside in the sense of moving down towards the ground. (I guess they might actually mean subside as the plume does spread downwards. It's slightly curious that nothing mixes up to higher heights within the troposphere. But this is likely to reflect this case only rather than generic behaviour.)

Changed to "preserve the plume's coherent structure". For "subside" we do mean moving down.

Page 3, lines 13-18: I think '(native)' means the resolution of the driving flow. If so then it would be good to say what was done at lower resolutions – hopefully the driving flow was smoothed rather than applying one grid point value over a larger area where it may be unrepresentative.

The wind field was regridded from 0.25°×0.3125° to lower resolutions using conservative regridding. This is now clarified.

Also I'm not sure what a '2-D horizontal plume' is in this context. Does this mean concentration is independent of height and depends only on x and y?

The simulation was performed in a 2D grid. (main text rephrased)

Eqns 19-21: While this is indicative, it's based on the idea of a coherent smoothly varying plume. In reality, once the plume is broken up into filaments with a lot of small scale structure, the second order derivatives will be dominated by the small scales, not the large scales which are characterised by L and H.

This is now recognized in the text following equation (21).

Page 8: This argument would not work if one was considering 3-D isotropic turbulence. Here one would expect k = 2 (or at least the power of $\Delta x$ would be twice the power of $\Delta z$ – both would have a CFL related increase), but it doesn't make sense to make $\Delta z$ smaller (or bigger) than $\Delta x$. Why does the argument go wrong for this case? One explanation for this is that perhaps one should replace (23) with D = max(A$\Delta x$,B$\Delta z$), consistent with page 8 lines 29-30, or replace it with D = A$\Delta x$ + A$\Delta y$ + B$\Delta z$, treating x, y, z on an equal footing. I think the authors either need to use one of these more balanced forms of D (with minor changes in the conclusions) or provide more justification for their choice.

The case for isotropic turbulence is now mentioned at the end of Section 4 and in the conclusion section.

Page 10, line 27 to page 11, line 3: It might be better to initially leave the description of $\lambda$ in words and not introduce $\partial u/\partial x$ (see fig 2 caption too) only to redefine it in (28). More importantly, stretching is not in general aligned with the local flow. One can add a uniform flow (Galilean transformation) in any direction without altering the stretching properties of the flow. The values are only used qualitatively however, so it's probably OK to use this as an indication of stretching.

Corrected.

Technical corrections:

Page 9, line 27: Should 'initialize' be 'run' or 'calculate'?

Corrected.

Page 11, line 8: 'Large scale vertical wind speeds' might be better. The values are too small for convective updrafts or boundary layer turbulence.

Corrected.

Fig 7 caption: 'straight lines' should be 'solid lines'.

Corrected.

**Reviewer #2**

The aim of this paper is to determine the horizontal and vertical resolution required for accurate free tropospheric plume transport. This topic is of interest and the analysis in this paper is presented clearly and precisely. The quality of the writing is excellent and there are therefore few specific comments. I recommend the paper for publication in ACPD with minor corrections.

General Comments

1. The paper deals specifically with free tropospheric tracer transport and situations in which the vertical velocities are small (order 1cm s-1). Do the results hold for different situations in which vertical velocities are larger? For example, in a convective boundary layer, A and B (eq 24) may be similar so B/A is much smaller than 500.

We now specifically discuss the difference with the convective boundary layer in the conclusion section (page 16, ~line 21).

2. In the abstract the authors claim that 'the local surface pollution influence from the subsiding plume on intercontinental scales is also considerably increased', but on p12 say that this result is just implied, as the fV3 dynamical core does not include boundary layer physics. Can the authors justify the term 'considerably' increased as they have no quantitative estimate of this potential increase?

We have removed "considerably"

3. Throughout the paper the authors refer to typical horizontal plume scales of 1000km. Given that in the real atmosphere plumes sizes cover a whole spectrum how sensitive are the results to the initial plume size? The authors present sensitivity results for plumes of different initial vertical thicknesses but not different initial horizontal spreads.

We also conducted sensitivity simulations with different horizontal extents and found that this did not affect the results. This is now stated in section 4.3.

On a similar topic of variable resolution, I was surprised that there was no discussion of in the introduction.

Adaptive mesh refinement is now mentioned at the end of the conclusion section (page 17, ~line 10).

Specific Comments

1. Page 2 line 6: the authors use 'thick plumes of Asian ozone pollution', do they refer to optical or geometric thickness here?

We have deleted 'thick'

2. Page 4 line 14: This sentence is a little difficult to follow. What is 'positive'?

Text has been clarified.

3. Page 5 line 21: CFL acronym should be defined.

Now defined.

4. Page 6 line 5: 'This explains why', what is 'this'? Are the authors referring to equation 14?

Changed to "Eq. (14)".

5. Page 7 line 14 and elsewhere: The authors often refer to 'reviewed above', 'will be discussed', 'described below' etc. Please can the specific section numbers be used in these places?

All corrected.

6. Page 13 line 4: 'S' is not defined.

Corrected

7. Page 14 line 6: Why is the z underlined here?

Typo, corrected

8. Page 30 figure B1: A difference plot might help to highlight deviations from the initial conditions due to numerical error more clearly.

Added.

**Reviewer #3**

This paper discusses the importance of vertical resolution relative to the horizontal resolution. The authors claim that there exists an optimal grid resolution ratio of 1000 between horizontal and vertical resolution.

The statement that the often used approach of increasing the horizontal resolution alone is insufficient for an optimal solution has been made before, also by the same authors. Also, as the authors themselves point out, optimal grid resolution ratios not too dissimilar to what is stated in the manuscript have been suggested before. This study is extension to previous work in that it aims to theoretically derive an optimal grid resolution ratio and to support the findings by performing idealised simulations with the GFDL-FV3 global dynamical core. Though not ground breaking the manuscript could be published after revisions as outlined below.

The paper makes very bold statements that are not widely valid and applicable since the findings are much more model, case and resolution dependent. This needs to be acknowledged and discussed more widely.

In response to the reviewer's concerns we have revised the manuscript in many places to provide a more even-handed assessment of the findings. Please refer to various comments below regarding the convection-resolving regime, the physical diffusion, etc.

For increasingly higher spatial resolutions that approach convection permitting or even convection resolving scales, an aspect ratio of 1000 wouldn't make much sense since the resolved dynamical features become more isotropic compared to a model where most of the vertical motions is on the subgrid scale.

Good point. We have added a paragraph to that effect at the end of Section 4.3 (page 15, ~line 30).

For a plume with an aspect ratio of 1000 (and flow ratios in the same order) as used here it is not surprising that a grid with the same aspect ratio provides an optimal solution.

There is no reason why $(\Delta x/\Delta z)_{opt}$ would scale as $L/H$ – see equation (21), it doesn't. We would rather not comment on this coincidence to avoid confusing the reader.

A validation with a real event would be limited by an incomplete knowledge of the atmospheric state. Only with a perfect knowledge of the atmospheric flow that drives the tracer transport the presented argument holds.

We can still diagnose the accuracy of the numerical scheme even without a perfect knowledge of the atmospheric flow. For pure advection, the VMR and entropy must be conserved, even if under highly

divergent flow (Rastigejev et al., 2010). Indeed, there is physical diffusion in the real atmosphere, thus the VMR and entropy cannot be preserved indefinitely. However, D'Isidoro et al., (2010) found that numerical diffusion dominates on the scales of global models. We added more discussions on this in Introduction (page 3, line 25~30).

Advection is not the only time step limiting process. Gravity waves, for example, pose a much stronger constrain. Since in global models the vertical resolution is higher than the vertical resolution (even if smaller than an aspect ratio of 1000) the numerical treatment is usually not the same for the vertical and the horizontal. GFDL-FV3 is an extreme example since it is not even Eulerian in the vertical. Although this is mentioned in section 3 this should already be mentioned at the beginning of section 2 since GFDL-FV3 is already discussed in section 1. Even then GFDL-FV3 doesn't fit the description of an Eulerian chemical transport model for which the theoretical analysis is performed in section 2. GFDL-FV3 is a full dynamical core with tracer transport so that the analysis in section 2 only applies to this model if aspects of predicting the flow are ignored. However, a more accurate prediction of the flow with increasing horizontal and/or vertical resolution is key for the interpretation of results in section 4. Appendix B is important but it shows similar numerical diffusion properties only for the transport, not for the dynamical core as a whole if I understand correctly.

We added more discussions on vertically Lagrangian vs. Eulerian at the end of Appendix B.

Lauritzen et al., (2010) compared dynamics simulations in Eulerian models and a vertically Lagrangian model (CAM-FV, which uses the same vertical scheme as FV3) and got highly consistent results. Thus we expect FV3 also has a similar dynamics property as common Eulerian models.

Volume mixing ratio is usually defined as tracer mass per unit volume and is thus not preserved in a non-divergent flow as stated. However, mass mixing ratio (the tracer mass per total mass) is. According to the figure labels, the authors define volume mixing ratio as tracer volume per total volume. This is only equivalent to mass mixing ratio for gaseous tracers, not for particulate tracers, like aerosols. To my knowledge, GFDL-FV3 solves for the tracer mass per unit dry air.

Volume mixing ratio is actually the same thing as molar mixing ratio for gases (moles ~ volume). It is preserved in divergent flow. To avoid confusion and extend generality to aerosols we changed the terminology to simply refer to 'mixing ratio' and define it in the intro as mass of chemical per mass of air.

I would have expected that section 4 confirms the optimal grid resolution ratio of 1000 but it does not. Since section 4 only focuses on error metrics for the tracer transport without including computer time, unsurprisingly, the case with the highest horizontal and vertical resolution produces the best results. At the highest resolution the aspect ratio is below 1000 (dx/dz=25km/80m=312.5) so that simulations with higher vertical resolution would be needed to be able to prove the point of section 2. Only for C48 do the performed simulations significantly exceed an aspect ratio of 1000. Simulations with higher vertical resolution at higher horizontal resolution are needed to prove a relationship based on equation (25).

We diagnose the optimal $\Delta x/\Delta z$ under computational constraints using the contour plots in Fig. 1 and 7, and the two plots are consistent. It is impossible to define the optimal $\Delta x/\Delta z$ without considering computational constraint. According to Eq. (12), increasing resolution (no matter in $\Delta x$ or $\Delta z$) will always lead to smaller numerical diffusion (although the benefit can be marginal if the other dimension is limiting). Thus, the highest resolution simulation (C384L160 here) will always have the best preservation of the plume, but its aspect ratio (312.5 here) is not necessarily the optimal ratio. Consider an extreme case: $(\Delta x, \Delta z) = (0.1$ km, $1$ km) will produce smaller numerical diffusion than $(\Delta x, \Delta z)=(10$ km, $1$ km). However, $(\Delta x, \Delta z) = (0.1$ km, $1$ km), or $\Delta x/\Delta z=0.1$, is considered as a bad configuration because the horizontal resolution is unnecessarily high and the error is limited by vertical resolution. In other words,

($\Delta x$, $\Delta z$) = (0.1 km, 1 km) is bad because it wastes too many computational resources on the horizontal resolution.

The initial conditions chosen for the test case are discontinuous at the plume edges. No matter what the spatial resolution, the grid well never resolve the transition between inside and outside plume. Thus, this test case cannot converge. It should still be possible to test the relationship for an optimal grid ratio as derived in section 2 but it would have been better to consider a test case that can numerically converge.

Traditional error convergence test can only be done in analytical flows, such as solid body rotation or idealized deformational flow, where the true analytical solution is known (Kent et al., 2014). In this work we focus on realistic/turbulent flows to address the issue raised by Rastigejev et al., (2010), and the analytical solution to our problem is unknown.

Instead, we choose the preservation of VMR and entropy as the diagnostics of simulation accuracy. With the entropy diagnostic, one might define the convergence of numerical error as the exactly preservation of entropy (Figure 5 in Lauritzen and Thuburn, 2012). However, in our highly challenging test with strongly stretched flow, the entropy still increases by a factor of 20 even at the highest resolution (C384L160). Thus, it is impossible to achieve error convergence, even if the plume is smooth at edges.

Thank you again for considering this work for *Atmospheric Chemistry and Physics*. We would like to also thank the reviewers for their time and their comments. The changes we have made in response to their concerns are highlighted in an attached copy of the manuscript. Additions are highlighted in blue, and deletions in red.

Sincerely,
Jiawei Zhuang

[revised manuscript text omitted]

---

## Referee Report (RR1)

Report on revised version of 'The importance of vertical resolution in the free troposphere for modeling intercontinental plumes' by Zhuang et al.

The revisions to the paper are welcome. The addition of the reference to D'Isidoro et al (2010) and associated discussion (p 3), as well as the discussion of the scope of the work (specifically the exclusion of convection and boundary layer turbulence), helps to set a better context and perspective for the work.

I have a remaining general comment on perspective, a number of minor comments, and one important comment concerning possible errors in the figures. While it would be nice to address all these issues, I think the 'important comment' is the only comment that's essential to address in the event that the authors consider, but disagree with, the other comments.

General comments:

I think there remain a few places where the perspective could be further improved. For example the authors say (p 13, line 31-32) 'the best possible simulation would preserve the entropy'. If, in the true solution, there are a lot of small filamentary structures on small scales which are much smaller than the grid scale (as seems likely at large times), then one has a choice between an entropy conserving solution with the filamentary structures being resolved and so on the wrong scale, or a solution which smoothes these small scales out and increases the entropy. It seems clear that the second of these, which does not preserve entropy, is to be preferred. One could argue that the authors' statement is correct if they mean the best possible simulation at any cost, but even this is doubtful because in reality there is molecular diffusion which will act on the smallest scales (as acknowledged in the discussion associated with the D'Isidoro et al reference). Note I'm not saying that reducing the increase in entropy a bit isn't a good thing (the increase may be too big due to numerical errors), but I am saying that 'the best possible simulation would preserve the entropy' is going too far.

I note that the Methven and Hoskins situation (mentioned in the authors' response), where small scales in the tracer field are generated by a smooth large scale wind (with chaotic trajectories), is in effect the situation I discussed previously in relation to Batchelor's $k^{-1}$ viscous-convective regime. If the smaller scales in the velocity field are present in reality (i.e. are not absent due to a very large viscosity and large Prandtl number), then not including them in the simulation is likely to lead to too much, not too little, small scale structure in the scalar field. This doesn't require a response but I mention it as it provides more context for some of my previous comments.

Important specific comment:

It looks from figure 6 as though none of the C48 cases have entropy below 19 on day 8. However figure 7 shows a case (C48,L160) with entropy of about 18 and another between 18 and 19. Also it looks from figure 6 like some of the L20 cases have entropy below 19 on day 8, but figure 17 has them all above 19. Have I got this right? It's easy to get it wrong with the C and L numbers increasing as $\Delta x$ and $\Delta z$ decrease. If I'm right, have the authors got the figure 7 entropy the wrong way round, or have they got the figure 6 plots

the wrong way round, or is it a consequence of the contour plotting program doing some smoothing? It's harder to see, but the right hand plots in figure 6 also look inconsistent with figure 7. I think this needs to be resolved before publication.

Minor specific comments:

Page 11, lines 28-30: This discussion of convection and boundary layer turbulence is welcome. However one can have plumes which remain low in the atmosphere (even in the boundary layer) and plumes can be affected by convection at any point in their transport. Hence it is not the case that convection and boundary layer turbulence are only relevant as part of 'plume initialization and termination processes'. I think it would be fine to simply say that convection and boundary layer turbulence are out of scope.

Page 12, line 5: It might be useful to say something about why the plume subsidence is typical of observations, although this is a side issue to the main topic of the paper and so not essential. It's not clear what the authors have in mind. On average vertical velocities must be zero, but this might not be so in particular parts of the world. The vertical velocities may be highly skewed, with an extreme case being deep convection and compensating subsidence, leading to mostly slow subsidence but occasional rapid uplift. Finally a plume released high up is likely to end up lower. The Crawford et al 2004 reference only appears (based on a very quick look) to argue for subsidence as a result of the particular flow regime being considered. Possibly the authors mean that the subsidence seen is a 'typical rate of subsidence for subsiding plumes', not that subsidence is typical?

A comment on Reviewer 3's last comment and the authors' response: I agree with reviewer 3 that it would be interesting to do this test with the initial plume having smooth edges, although this could be regarded as a separate project. I'm not sure how to interpret the authors' response. Perhaps they have done the test and the entropy increases by a factor of about 20 at the highest resolution – if so it might be of interest to report this in the paper.

---

## Referee Report (RR2)

Review of the revised manuscript "The importance of vertical resolution in the free troposphere for modelling intercontinental plumes" by Zhuang et al.

The authors satisfactorily addressed most of my comments.

I am still not fully convinced that the numerical plume simulations in section 4 actually prove and confirm the theoretical analysis of section 2. The authors also claim that the consistency between Fig. 1 and Fig 7. would confirm the theoretical analysis by the numerical simulations. However, all lines shown in Fig 7. are based on the scaling arguments from section 2. No actual run time (computer resource) information is included. From the information given in the manuscript it seems that since the time step is adjusted according to the horizontal resolution a scaling following the red lines in Fig. 7 should be expected. This should be made more explicit and a short paragraph about the actual scaling of computing time should be added. Additional factors/penalties not considered here (for example computational overhead due to parallelisation) might lead to deviation from the expected scaling.

Minor correction:
Page 16, line 21 should read lower L/H and smaller (not larger) dx/dz ratios.

---

## Author Response (AR2)

Dear Editor,

Thank you again for arranging the careful reviews. We would like to also thank the reviewers again for their comments. Our revised manuscript is enclosed; please find below our responses (in blue) to each of the comments made by the reviewers.

**Reviewer #1**

General comments:
I think there remain a few places where the perspective could be further improved. For example the authors say (p 13, line 31-32) 'the best possible simulation would preserve the entropy'. If, in the true solution, there are a lot of small filamentary structures on small scales which are much smaller than the grid scale (as seems likely at large times), then one has a choice between an entropy conserving solution with the filamentary structures being resolved and so on the wrong scale, or a solution which smoothes these small scales out and increases the entropy. It seems clear that the second of these, which does not preserve entropy, is to be preferred. One could argue that the authors' statement is correct if they mean the best possible simulation at any cost, but even this is doubtful because in reality there is molecular diffusion which will act on the smallest scales (as acknowledged in the discussion associated with the D'Isidoro et al reference). Note I'm not saying that reducing the increase in entropy a bit isn't a good thing (the increase may be too big due to numerical errors), but I am saying that 'the best possible simulation would preserve the entropy' is going too far.

Good point. The main text is rephrased (page 13, line 32).

I note that the Methven and Hoskins situation (mentioned in the authors' response), where small scales in the tracer field are generated by a smooth large scale wind (with chaotic trajectories), is in effect the situation I discussed previously in relation to Batchelor's $k-1$ viscous-convective regime. If the smaller scales in the velocity field are present in reality (i.e. are not absent due to a very large viscosity and large Prandtl number), then not including them in the simulation is likely to lead to too much, not too little, small scale structure in the scalar field. This doesn't require a response but I mention it as it provides more context for some of my previous comments.

Important specific comment:
It looks from figure 6 as though none of the C48 cases have entropy below 19 on day 8. However figure 7 shows a case (C48,L160) with entropy of about 18 and another between 18 and 19. Also it looks from figure 6 like some of the L20 cases have entropy below 19 on day 8, but figure 17 has them all above 19. Have I got this right? It's easy to get it wrong with the C and L numbers increasing as Δx and Δz decrease. If I'm right, have the authors got the figure 7 entropy the wrong way round, or have they got the figure 6 plots the wrong way round, or is it a consequence of the contour plotting program doing some smoothing? It's harder to see, but the right hand plots in figure 6 also look inconsistent with figure 7. I think this needs to be resolved before publication.

Figure 7, the contour plot, was incorrectly transposed. Thanks very much for finding this mistake. Correcting the plot changes our estimate of optimal $\Delta x/\Delta z$ from ~1000 to ~500. It is still consistent with the theoretical analysis in Section 2, which only provides an order-of-magnitude estimate. It does not affect other parts of discussions.

The bug fix to the plotting code is recorded in more details in our code repository:
https://github.com/JiaweiZhuang/FV3_util/commit/afdb03f717871e9aa3f56a95f7ff498c57b4da09

Minor specific comments:

Page 11, lines 28-30: This discussion of convection and boundary layer turbulence is welcome. However one can have plumes which remain low in the atmosphere (even in the boundary layer) and plumes can be affected by convection at any point in their transport. Hence it is not the case that convection and boundary layer turbulence are only relevant as part of 'plume initialization and termination processes'. I think it would be fine to simply say that convection and boundary layer turbulence are out of scope.

Corrected.

Page 12, line 5: It might be useful to say something about why the plume subsidence is typical of observations, although this is a side issue to the main topic of the paper and so not essential. It's not clear what the authors have in mind. On average vertical velocities must be zero, but this might not be so in particular parts of the world. The vertical velocities may be highly skewed, with an extreme case being deep convection and compensating subsidence, leading to mostly slow subsidence but occasional rapid uplift. Finally a plume released high up is likely to end up lower. The Crawford et al 2004 reference only appears (based on a very quick look) to argue for subsidence as a result of the particular flow regime being considered. Possibly the authors mean that the subsidence seen is a 'typical rate of subsidence for subsiding plumes', not that subsidence is typical?

We do mean that the rate of subsidence is typical of observations (main text corrected).

A comment on Reviewer 3's last comment and the authors' response: I agree with reviewer 3 that it would be interesting to do this test with the initial plume having smooth edges, although this could be regarded as a separate project. I'm not sure how to interpret the authors' response. Perhaps they have done the test and the entropy increases by a factor of about 20 at the highest resolution – if so it might be of interest to report this in the paper.

We have effectively done the test because the initially-sharp plumes quickly become smooth at boundaries during advection (Fig. 3). Due to the buffering effect of smooth boundaries, the dilution of plumes does get slower over time (Fig. 4 and 6), but the diffusion keeps going on. There is no sign that the entropy will stop increasing (Fig. 6). We now comment on this in the discussion of Figure 6 in Section 4.2 (page 14, line 11).

**Reviewer #3**

I am still not fully convinced that the numerical plume simulations in section 4 actually prove and confirm the theoretical analysis of section 2. The authors also claim that the consistency between Fig. 1 and Fig 7. would confirm the theoretical analysis by the numerical simulations. However, all lines shown in Fig 7. are based on the scaling arguments from section 2. No actual run time (computer resource) information is included. From the information given in the manuscript it seems that since the time step is adjusted according to the horizontal resolution a scaling following the red lines in Fig. 7 should be expected. This should be made more explicit and a short paragraph about the actual scaling of computing time should be added. Additional factors/penalties not considered here (for example computational overhead due to parallelisation) might lead to deviation from the expected scaling.

GFDL-FV3 is designed for efficient parallel execution and has little overhead. It shows near-linear scalability to ~1000 cores at C384 resolution (Chapter 5.3 of Putman, 2007).

The actual computational costs (cores × hours) of our simulations follow the $\Delta z \Delta x^3 = P$ curve. This is now mentioned on page 14, line 23. Using the theoretical line ($\Delta z \Delta x^3 = P$) allows us to more easily estimate the optimal point. The real timing results are only available at discrete points and would require interpolation (like the maximum mixing ratio and entropy).

We did not present detailed timing information in the main text, since they were not set up to follow the standard strong-scaling or weak-scaling tests. The number of cores were chosen for efficient execution and resource allocation, not for rigorous scaling analysis.

We provide the timing results in Table 1 below, in case the reviewer is interested. The actual CPU time is consistent with the theoretical estimate of CPU time given by $1/(\Delta z \Delta x^3)$. C96 has a slightly better parallelization efficiency than C48 (C96L20 only uses 7 CPU hours, not 8), because both C48 and C96 use 24 cores while C96 has a lower communication-to-computation ratio.

| Wall time (seconds) | L20 | L40 | L80 | L160 |
|---|---|---|---|---|
| C48 | 30.62 | 61.96 | 125.11 | 264.08 |
| C96 | 216.21 | 434.18 | 858.10 | 1762.35 |
| C192 | 435.57 | 863.83 | 1770.67 | 3591.50 |
| C384 | 834.57 | 1726.19 | 3576.95 | 6798.91 |

| # of cores | L20 | L40 | L80 | L160 |
|---|---|---|---|---|
| C48 | 24 | 24 | 24 | 24 |
| C96 | 24 | 24 | 24 | 24 |
| C192 | 96 | 96 | 96 | 96 |
| C384 | 384 | 384 | 384 | 384 |

| Normalized CPU hours | L20 | L40 | L80 | L160 |
|---|---|---|---|---|
| C48 | 1.00 | 2.02 | 4.09 | 8.62 |
| C96 | 7.06 | 14.18 | 28.02 | 57.55 |
| C192 | 56.89 | 112.83 | 231.28 | 469.10 |
| C384 | 436.03 | 901.87 | 1868.82 | 3552.16 |

| Theoretical CPU hours | L20 | L40 | L80 | L160 |
|---|---|---|---|---|
| C48 | 1 | 2 | 4 | 8 |
| C96 | 8 | 16 | 32 | 64 |
| C192 | 64 | 128 | 256 | 512 |
| C384 | 512 | 1024 | 2048 | 4096 |

Table 1. Timing results of GFDL-FV3 model simulations. "Normalized CPU hours" are calculated by [wall time] × [number of cores], normalized by the C48L20 case. "Theoretical CPU hours" are calculated from $1/(\Delta z \Delta x^3)$, normalized by the C48L20 case.

The code for generating Table 1, as well as the raw log files containing the timing information, are available in our code repository for record:
https://github.com/JiaweiZhuang/FV3_util/tree/master/small_output/outputlog

Minor correction:
Page 16, line 21 should read lower L/H and smaller (not larger) dx/dz ratios.

In the boundary layer, models often have higher vertical resolution (small vertical grid spacing $\Delta z$), so the $\Delta x/\Delta z$ ratio should be larger.

**Reference**

Putman, W. M.: Development of the Finite-Volume Dynamical Core on the Cubed-Sphere, The Florida State University., 2007.

The changes we have made in response to their concerns are highlighted in an attached copy of the manuscript. Additions are highlighted in blue, and deletions in red. We have also updated the public code repository to reflect technical corrections (https://github.com/JiaweiZhuang/FV3_util/releases). The updated DOI is 10.5281/zenodo.1214605.

Sincerely,
Jiawei Zhuang

[revised manuscript text omitted]